# Discovery and characterization of H$_v$1-type proton channels in reef-building corals

**Gisela Rangel-Yescas[1], Cecilia Cervantes[1], Miguel A Cervantes-Rocha[1], Esteban Suárez-Delgado[1], Anastazia T Banaszak[2], Ernesto Maldonado[3], Ian Scott Ramsey[4], Tamara Rosenbaum[5], Leon D Islas[1]\***

[1]Departmento de Fisiología, Facultad of Medicina, Universidad Nacional Autónoma de México, Mexico City, Mexico; [2]Unidad Académica de Sistemas Arrecifales, Instituto de Ciencias del Mar y Limnología, Universidad Nacional Autónoma de México, Puerto Morelos, Mexico; [3]EvoDevo Research Group, Unidad Académica de Sistemas Arrecifales, Instituto de Ciencias del Mar y Limnología, Universidad Nacional Autónoma de México, Puerto Morelos, Mexico; [4]Department of Physiology and Biophysics, School of Medicine, Virginia Commonwealth University, Richmond, United States; [5]Departmento de Neurociencia Cognitiva, Instituto de Fisiología Celular, Universidad Nacional Autónoma de México, Mexico City, Mexico

**Abstract** Voltage-dependent proton-permeable channels are membrane proteins mediating a number of important physiological functions. Here we report the presence of a gene encoding H$_v$1 voltage-dependent, proton-permeable channels in two species of reef-building corals. We performed a characterization of their biophysical properties and found that these channels are fast-activating and modulated by the pH gradient in a distinct manner. The biophysical properties of these novel channels make them interesting model systems. We have also developed an allosteric gating model that provides mechanistic insight into the modulation of voltage-dependence by protons. This work also represents the first functional characterization of any ion channel in scleractinian corals. We discuss the implications of the presence of these channels in the membranes of coral cells in the calcification and pH-regulation processes and possible consequences of ocean acidification related to the function of these channels.

**\*For correspondence:**
leon.islas@gmail.com

## Introduction

Scleractinian or stony corals are organisms in the phylum Cnidaria that deposit calcium carbonate (CaCO$_3$) in the form of aragonite to build an exoskeleton. Stony corals are the main calcifying organisms responsible for the construction of coral reefs, which are major ecosystems hosting numerous and diverse organisms. Coral reefs also act as natural barriers from strong ocean currents, waves, and tropical storms, providing coastal protection. This protection centers on the ability of scleractinian corals to produce enough CaCO$_3$. The increase in atmospheric CO$_2$ concentrations as a result of human activity poses threats to coral-reef-building organisms due to rising sea surface temperatures (*Hoegh-Guldberg, 1999*) and because CO$_2$ is taken up by the ocean, dangerously lowering the pH of the sea water (*Caldeira and Wickett, 2003*).

It is known that precipitation of the aragonitic form of calcium carbonate is facilitated at elevated pH values, at very low concentrations of protons. Calcification by scleractinian corals is a process that has been shown to be modulated by the pH of the solution in which calcium carbonate is precipitated (*Allemand et al., 2011*). To this end, corals produce a specialized compartment between the ectoderm and the external substrate or skeleton called calicoblastic compartment, which

contains a fluid derived from the surrounding sea water. The composition of this calicoblastic fluid or liquor is strictly regulated by the coral to maintain both an elevated pH, often close to one unit higher than the surrounding sea water, and an increased concentration of $Ca^{2+}$ and carbonates. The molecular details of pH regulation in the calicoblastic fluid are not understood completely. Involvement of proton pumps has been postulated and is likely to be part of proton transport in corals. Both P-type and V-type hydrogen pumps are present in coral transcriptomes and are known to play roles in the physiology of coral-algal symbiosis (*Tresguerres et al., 2017*). V-type $H^+$-ATPases have also been shown to be involved in calcification in foraminifera (*Toyofuku et al., 2017*). If a proton pump is involved in lowering proton concentration in the calicoblastic fluid to maintain high calcification rates, protons will be transported to the cytoplasm of the ectodermal cells that constitute the calicoblastic epithelium, producing a profound acidification of the cytoplasmic pH ($pH_i$). Although measurements of the $pH_i$ in corals indicate values of 7.13–7.4 (*Venn et al., 2009*), it is unknown how coral cells regulate $pH_i$. Thus, an efficient pH-regulatory mechanism is to be expected to be present in corals. We hypothesized that proton channels might be fundamental to this physiological process and also required for calcification in hard corals.

Although a number of studies have delineated the physiological roles of $H_v1$ voltage-gated proton channels in vertebrate cells (*DeCoursey, 2013*), less is known about their role in invertebrates. These channels are potential mediators in processes that are critically dependent on proton homeostasis. As an example, they have been shown to be involved in regulating the synthesis of the calcium carbonate skeleton in coccolithophores, calcifying unicellular phytoplankton (*Taylor et al., 2011*).

The range of voltages over which channel activation occurs is strongly modulated by the transmembrane proton gradient, characterized by $\Delta pH = pH_o\text{-}pH_i$, that is, the difference between the external and internal pH. In the majority of known $H_v1$ channels, the voltage at which half of the channels are activated, the $V_{0.5}$ or the apparent threshold for channel opening ($V_{Thr}$), shifts by roughly 40 mV per unit of $\Delta pH$. Thus, the pH gradient strongly biases the voltage-independent free energy of channel activation (*Cherny et al., 1995*). With few exceptions, channel activation occurs at voltages that are more positive than the reversal potential for protons, implying that protons are always flowing outward under steady-state conditions. The fact that most $H_v1s$ mediate outward currents is the reason these channels are mostly involved in reversing intracellular acidification or producing voltage-dependent cytoplasmic alkalization (*Lishko and Kirichok, 2010*; *DeCoursey, 2013*).

Here we report the presence of genes encoding $H_v1$ channels in two species of reef-building corals. We cloned and characterized the biophysical properties of these channels in an expression system using patch-clamp electrophysiology. The demonstration of the presence of voltage-gated proton channels in corals is an initial step to a deeper understanding of coral calcification and its dysregulation under ocean acidification conditions. We show that some of the coral $H_v1$'s biophysical properties are different from other known proton channels, and this behavior makes them interesting models to try to understand some basic biophysical mechanisms in these channels. To explain this behavior, we developed a novel activation model to describe voltage- and pH-dependent gating that has general applicability to $H_v1$ channels.

## Results

Ion channels have not been characterized in corals. Here, we have initiated their study by searching the transcriptome of the Indo-Pacific coral *Acropora millepora* (*Moya et al., 2012*) for sequences coding for putative voltage-sensing residues present in canonical $H_v1$ channels with the form RxxRxxRIx, which corresponds to the S4 segment of $H_v1$ channels and is also found in other voltage-sensitive membrane proteins. Blast searches detected four sequences that seem to correspond to a gene encoding the $H_v1$ voltage-activated proton-selective ion channel (*Ramsey et al., 2006*; *Sasaki et al., 2006*). *A. millepora* is one of the most widely studied species of scleractinian corals and is well represented in the commercial coral trade (*Cleves et al., 2018*; *Ying et al., 2019*). We proceeded to clone this gene from a small specimen of *A. millepora* obtained from a local aquarium (Reef Services, Mexico City). As indicated in the 'Materials and methods' section, total RNA was extracted from tissue and mRNA was retrotranscribed to obtain complementary DNA (cDNA). We managed to obtain a full-length clone and refer to this sequence as $AmH_v1$ or $H_v1$-type proton channel of *A. millepora*.

We were interested in knowing if the same gene is present in a closely related species from the Caribbean Sea. Thus, we used the same primers to clone the H$_v$1 channel from *Acropora palmata*, a widespread coral in the same family, and which we call ApH$_v$1. The amino acid sequence is almost identical to AmH$_v$1 (*Figure 1—figure supplement 1A*); the greatest divergence is found between a few amino acid residues in the C-terminal region. This result suggests that despite the large biogeographic difference, these two genes have not diverged significantly. The ApH$_v$1 sequence also gives rise to fast-activating voltage-gated proton currents (*Figure 1—figure supplement 1B*).

The most diagnostic feature of the H$_v$1 protein is the sequence of the fourth transmembrane domain or S4, which contains three charged amino acids in a characteristic triplet repeat. The presence of these repeats in our sequence allowed us to initially identify our clone as an H$_v$1 channel. However, we decided to compare our sequence to those of several H$_v$1 orthologs. We selected a list of 130 H$_v$1 protein sequences that are well curated in the Gene Bank (https://www.ncbi.nlm.nih.gov/), representing several branches of the eukaryotes, from unicellular plants to mammals. As expected, the protein sequence of AmH$_v$1 has similarity to several other H$_v$1 genes from varied organisms (*Figure 1A*). The identity varies from 98%, when compared to other putative coral and anemone sequences, to less than 30%, when compared to plant and nematode sequences. In spite of this variability, the putative transmembrane domains of all these proteins show high conservation, and consensus sequence logos can detect the presence of highly conserved individual amino acid sequences that can be considered characteristic of H$_v$1 channels. *Figure 1B* compares these transmembrane domain consensus logos with our AmH$_v$1 sequence. It can be gleaned that AmH$_v$1 contains the highly conserved residues that form the voltage-sensing amino acid residues in S4 as well as their acidic pairs present in S2 and S3. The extracellular histidine residues involved in Zn$^{2+}$ coordination are also present. These results suggest that our sequence is that of a bona fide H$_v$1 voltage-sensing domain (VSD).

Apart from canonical voltage-gated channels, several other proteins contain VSDs. Examples are the voltage-sensing phosphatases like VSPs (*Iwasaki et al., 2008*) and TPTE and TPTE2 (*Halaszovich et al., 2012*) proteins (transmembrane proteins with tensin homology) and genes like TMEM266. These proteins are relevant to us since some TPTEs have been shown to also mediate proton currents and TMEM266 can be modulated by Zn$^{2+}$ (*Papp et al., 2019*). We compared the sequence of AmH$_v$1 with several orthologs of TPTEs and TMEM266. Although there is some similarity within transmembrane domains (*Figure 1—figure supplement 2*), the overall sequence comparison shows that AmH$_v$1 and these VSD-containing proteins are different.

As mentioned before, we performed a multiple sequence alignment with 130 H$_v$1 sequences. In *Figure 2*, we show the detailed sequence alignment of AmH$_v$1 with five of these sequences, which represent some of the best studied H$_v$1 genes. It can be seen that there is a high degree of identity, especially in the transmembrane domains. The least degree of conservation appears when comparing this sequence to the dinoflagellate *Karlodinium veneficum* H$_v$1 channel (*Figure 2A*). A search of available transcriptomes from several coral species allowed us to detect the presence of sequences that are found in H$_v$1 channels. This suggests that H$_v$1 proton channels might be found in many families of scleractinian corals (*Figure 2—figure supplement 1*), as has also been recently shown (*Capasso et al., 2021*).

Secondary-structure prediction suggests that AmH$_v$1 is a canonical H$_v$1 channel formed by a VSD with four transmembrane segments. The protein sequence was used for 3D modeling using the SWISS MODEL server (*Waterhouse et al., 2018*), which produced models based on the H$_v$1 chimera structure (*Takeshita et al., 2014*) and the Kv1.2 potassium channel VSD (*Long et al., 2005*). This structural model is shown in *Figure 2B*. The predicted model indicates a shortened N-terminal region, four transmembrane helices, and a long C-terminal helix.

Voltage-gated proton channels from *Ciona* (*Sasaki et al., 2006*) and humans (*Lee et al., 2008*) have been shown to express as dimers in the plasma membrane, and this dimeric form is understood to be the functional unit of these proton channels. The dimer is stabilized by a coiled-coil interaction mediated by an alpha helical C-terminal domain. As shown by the model in *Figure 2B*, AmH$_v$1 has a long C-terminal helix, which is predicted to engage in a coiled coil (Paircoil2; *McDonnell et al., 2006*). We calculated the probability per residue to form a coiled coil for all the C-terminal residues, both for human and AmH$_v$1 channels, using the program COILS (*Lupas et al., 1991*). *Figure 3A* shows that the coiled-coil probability for AmH$_v$1 C-terminus is at least as high or higher than that for hH$_v$1, an established dimer, strongly suggesting that coral H$_v$1s might also form dimers.

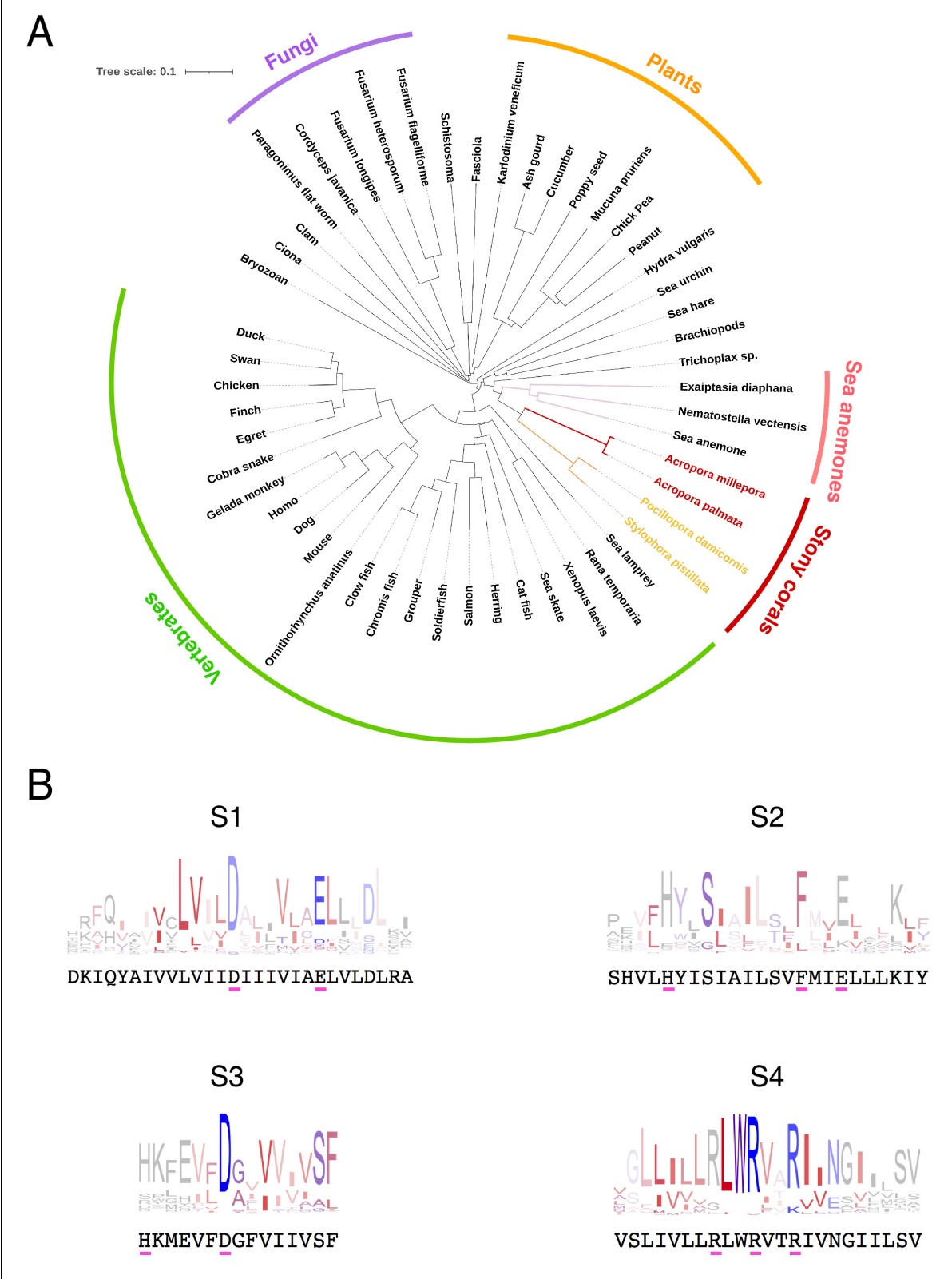

**Figure 1.** Conservation and phylogenetic relationships of $H_v1$ channels. (**A**) The tree obtained from a multiple sequence alignment from $H_v1$ channels in CLUSTAL-O. Highlighted in red and yellow are the branches containing coral $H_v1$ sequences. (**B**) Consensus logo sequences of transmembrane domains of $H_v1$ channels. The color code indicates the hydrophobicity of each residue, where blue indicates charged residues, red indicates non-polar residues, and other colors indicate either non-polar or charged residues with less conservation.

*Figure 1 continued on next page*

*Figure 1 continued*

The online version of this article includes the following source data and figure supplement(s) for figure 1:

**Source code 1.** Code for generating the tree in *Figure 1*.

**Figure supplement 1.** Some characteristics of H$_v$1 from *Acropora palmata*.

**Figure supplement 2.** Comparison of the sequence of AmH$_v$1 to other voltage-sensing proteins.

In order to study the oligomeric state of the coral H$_v$1, we performed FRET experiments with the AmH$_v$1 channel tagged with fluorescent proteins (FPs) as a FRET pair. *Figure 3B* shows that there is significant FRET efficiency between FP-tagged subunits, indicating a very close interaction between monomers. The measured apparent FRET efficiency vs the fluorescence intensity ratio can be fitted to a model where the subunits assemble as a dimer. From this fit, we can estimate a distance between fluorophores of ~60 Å, which is compatible with AmH$_v$1 being a dimer, at least in HEK293 cells.

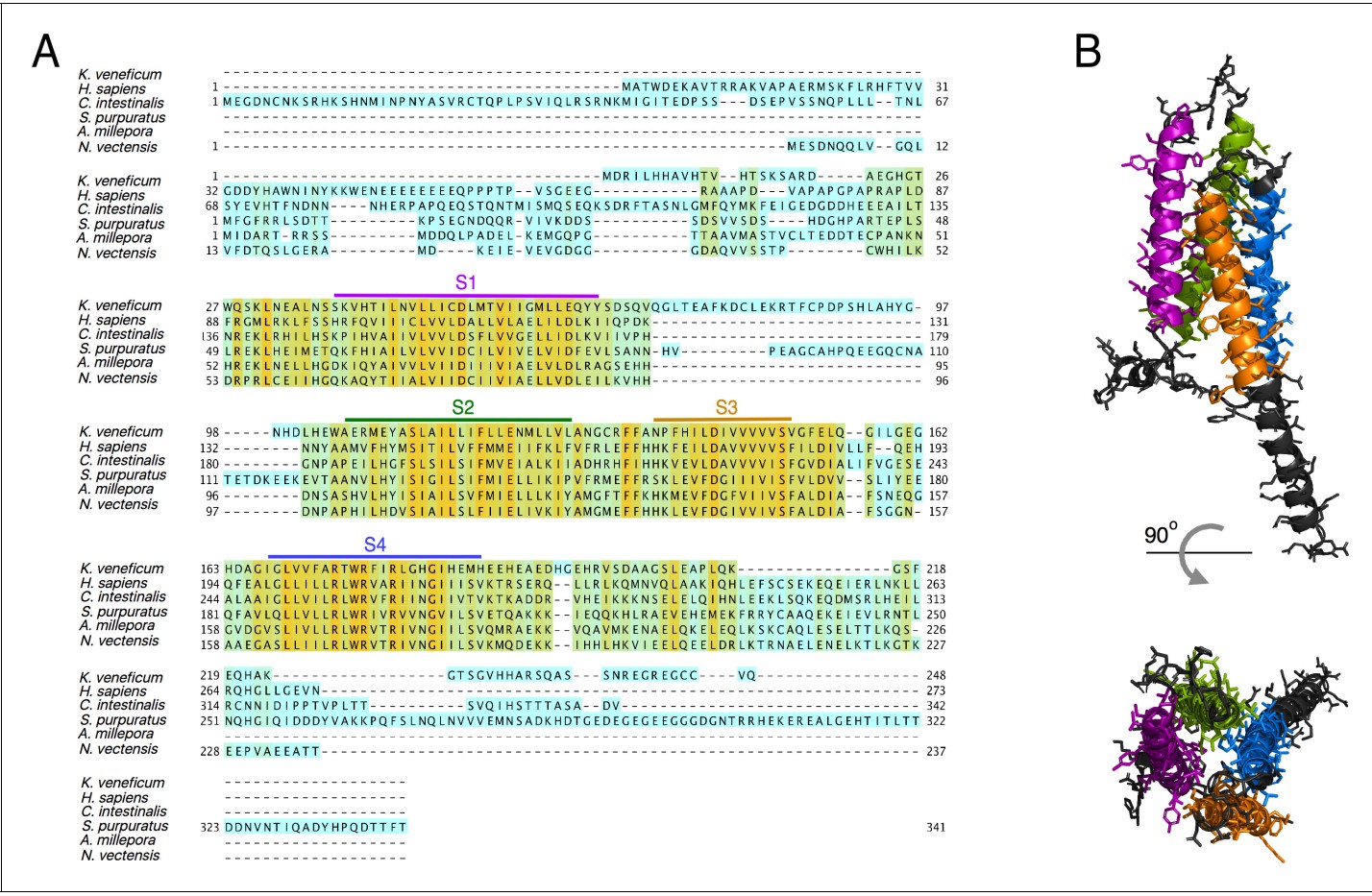

**Figure 2.** Protein sequence alignment of the AmH$_v$1 channel with selected H$_v$1s from other organisms. (**A**) Amino acid sequence alignment of *Acropora millepora* H$_v$1 (AmH$_v$1) with other known H$_v$1 orthologs provided by the CLUSTAL-O algorithm. The predicted transmembrane domains are shown by the colored horizontal lines and letters. The colors highlighting the sequence indicate sequence identity. Orange indicates identical amino acids, and cyan indicates no identity. (**B**) Predicted structural topology of AmH$_v$1. Transmembrane domains are colored to correspond with the sequences in (**A**). The top panel is the view parallel to the membrane while the bottom panel is the view from the top (extracellular) side.

The online version of this article includes the following figure supplement(s) for figure 2:

**Figure supplement 1.** Comparison of the AmH$_v$1 protein sequence with similar sequences found in other coral species.

## Functional expression of AmH$_v$1 voltage-dependence and kinetics

The cDNA of AmH$_v$1 was cloned in the pcDNA3 expression vector and transfected into HEK293 cells. Under whole-cell conditions, we recorded large voltage-dependent outward currents. *Figure 4A* shows a family of such currents. The data suggest that these currents were carried mostly by protons, since the reversal potential, measured from a tail current protocol, closely followed the equilibrium potential for protons, as given by the Nernst equation (*Figure 4B*).

The voltage-dependence of channel gating was estimated from a fit of the normalized conductance vs voltage (G-V) to *Equation 1*. The steepness of the curve corresponds to an apparent charge of ~2 e$_o$, comparable to other H$_v$1s under similar recording conditions (*Figure 4C*).

Interestingly, these channels seem to activate rapidly. This is apparent from the current traces, which approach a steady state within a few hundred ms (*Figure 4A*), as quantified in *Figure 4D*. *Equation 3* estimates two parameters, an activation time constant (τ) and a delay (δ). Both the time constant and the delay are similarly voltage-dependent at positive potentials. The existence of a delay in the time course implies that activation is a multiple-state process. The delay magnitude is

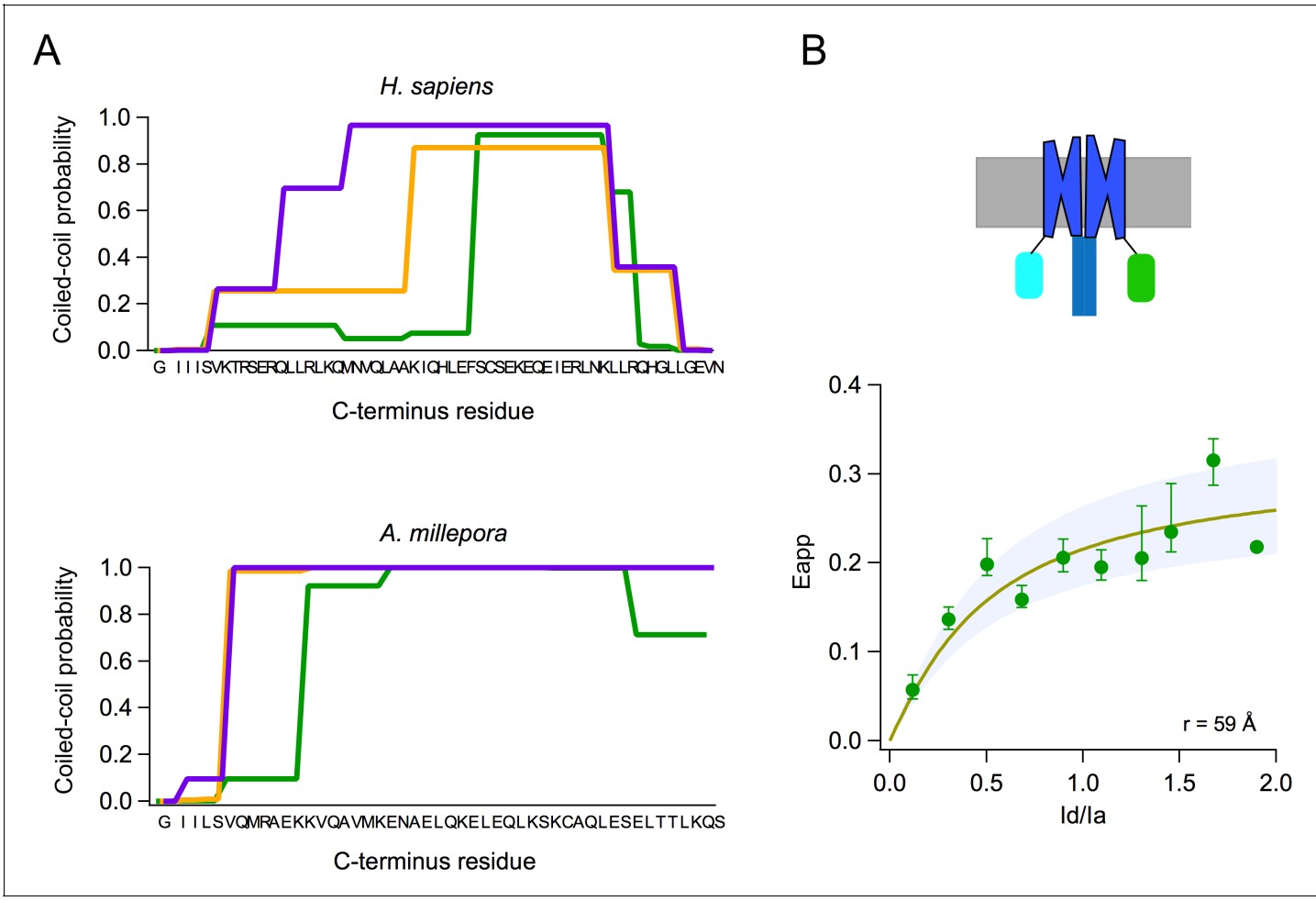

**Figure 3.** Subunits of the AmH$_v$1 channel associate to form dimers. (A) Probability of coiled-coil formation per amino acid residue of the C-terminus domain of hH$_v$1 (top) and *Acropora millepora* H$_v$1 (AmH$_v$1) (bottom). The different colors correspond to the three seven-residue windows used by the program to calculate the score. The sequence of the C-terminus is shown in the x-axis. (B) FRET measurement of dimer formation. The apparent FRET measured from 134 cells is plotted as a function of the ratio of donor to acceptor fluorescence (I$_d$/I$_a$). Shown is the average and sem for data in I$_d$/I$_a$ windows of 0.1. The continuous curve is the fit of the data to the prediction of a model that considers random assembly of donor- and acceptor-tagged subunits into a dimer. The separation between the FRET pairs in a dimer is ~60 Å, according to the model. The upper panel depicts a cartoon of the presumed fluorescent protein (FP)-tagged dimer in the membrane.

The online version of this article includes the following source data for figure 3:

**Source data 1.** Source data for *Figure 3*.

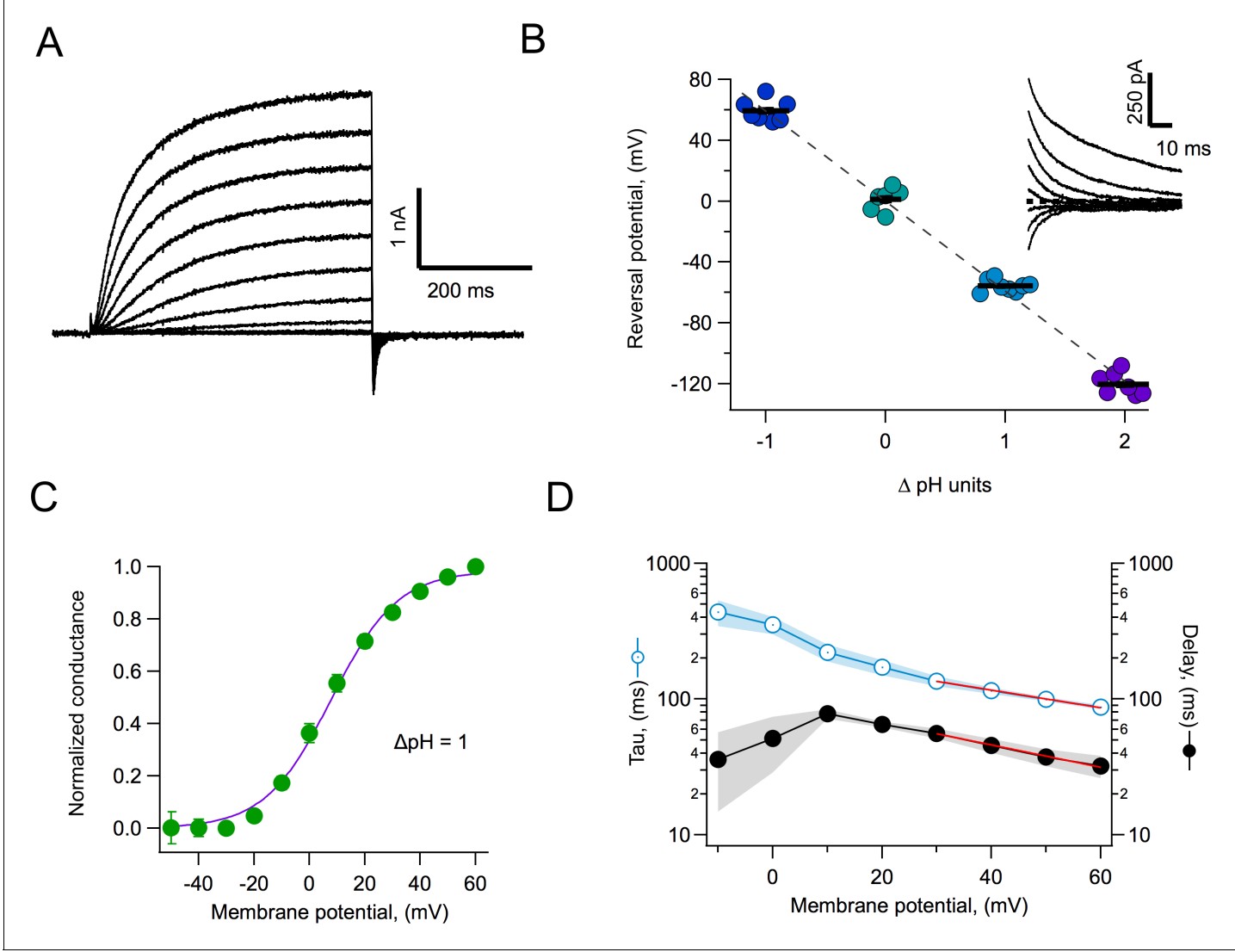

**Figure 4.** Proton currents mediated by AmHv1 expressed in HEK293 cells. (A) A typical proton current family elicited by depolarizing pulses from −50 to 60 mV in 10 mV intervals. The duration of the pulses is 500 ms. Linear current components have been subtracted. (B) Reversal potential of currents as a function of the pH gradient. Symbols are individual data and the black horizontal lines are the mean. The dotted line is the expected reversal potential as predicted by the Nernst equation. The inset shows a tail current family from which instantaneous IV curves were extracted to measure the reversal potential. Recordings shown in (A) and (B) were obtained in the whole-cell configuration. (C) Normalized conductance-voltage curve at ΔpH = 1. The purple curve is the fit to *Equation 1* with parameters $V_{0.5}$ = 7.85 mV and q = 2.09 $e_o$. Circles are the mean and error bars are the sem (n = 7). (D) Kinetic parameters of activation. Activation time constant and delay estimated from fits of current traces to *Equation 2*. Circles are the mean, and the sem is indicated by the shaded areas (n = 6). The voltage-dependence of the delay and tau of activation were estimated from a fit to *Equation 3*, which appears as the red curve. Parameters are $\delta(0)$ = 98.2 ms and $q_\delta$ = 0.47 $e_o$. The voltage-dependence parameters for tau are $\tau(0)$ = 212 ms and $q_\tau$ = 0.37 $e_o$.

The online version of this article includes the following source data for figure 4:

**Source data 1.** Source data for *Figure 4*.

smaller than the time constant at all voltages, which can be interpreted to mean that the rate-limiting step for opening comes late in the activation pathway (*Schoppa and Sigworth, 1998*).

## Comparison to human Hv1 channel properties

Human Hv1 is probably the best characterized of the voltage-gated proton channels (*Musset et al., 2008*); so we compared some of the properties of AmHv1 with hHv1. AmHv1 channels activate faster than their human counterpart. *Figure 5* compares the activation kinetics of these two channels under

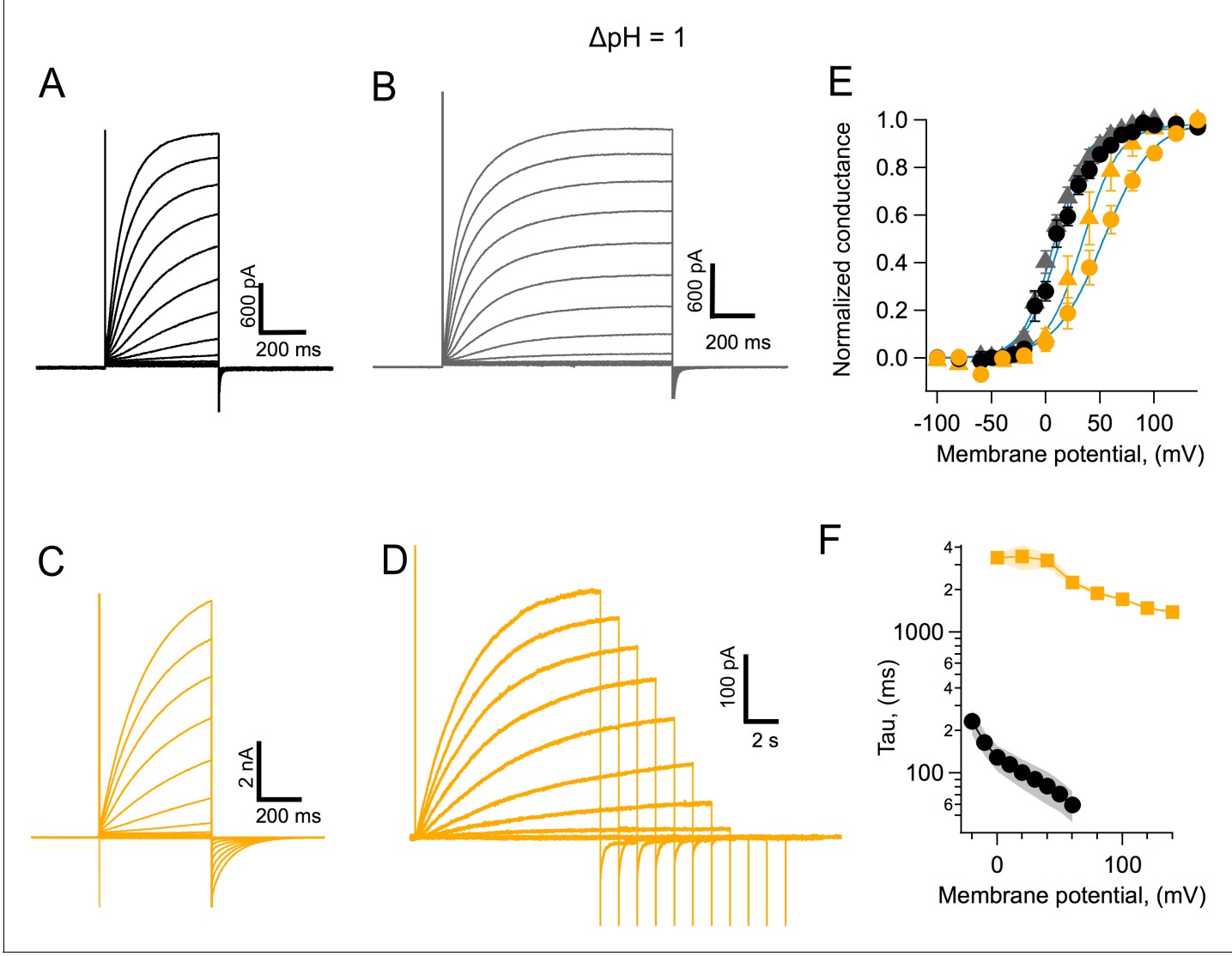

**Figure 5.** Coral H$_v$1 channels are faster and activate more readily than their human counterpart. (A) *Acropora millepora* H$_v$1 (AmH$_v$1) currents in response to voltage-clamp pulses from −100 to 120 mV and of 500 ms duration. (B) AmH$_v$1 currents in response to the same voltage-clamp pulses as in (A) but of a longer duration of 800 ms. (C) Currents through hH$_v$1 channels in response to voltage-clamp pulses from −100 to 120 mV of 500 ms duration, compared with (A). (D) hHv1 currents recorded with long pulses designed to reach the steady state. Pulses were shortened in duration as depolarizations became larger, in an effort to reduce intracellular proton depletion. Pulses are from −100 to 140 mV. Recordings shown in (A), (B), (C), and (D) were obtained in the whole-cell configuration. (E) Comparison of the conductance-voltage relationship for both channels for short- (circles) and long (triangles)-duration pulses. Black symbols are the mean G/G$_{max}$ for AmH$_v$1 and yellow symbols for hH$_v$1. The error bars are the sem (n = 3, for short pulses, both channels, and n = 4, for long pulses, both channels). The continuous blue curves are fits to *Equation 1*. The fitted parameters are AmH$_v$1, short pulses, q = 1.62 e$_o$, V$_{0.5}$ = 12.2 mV; AmH$_v$1, long pulses, q = 1.7 e$_o$, V$_{0.5}$ = 7.7 mV; hH$_v$1, short pulses, q = 1.11 e$_o$, V$_{0.5}$ = 53.1 mV; hH$_v$1, long pulses, q = 1.47 e$_o$, V$_{0.5}$ = 34.1 mV. (F) The activation time constant estimated from fits of currents elicited by long pulses to *Equation 2*. Squares are the mean for hH$_v$1 and circles, for AmH$_v$1. The shaded areas are the sem (n = 4, for both channels).

The online version of this article includes the following source data for figure 5:

**Source data 1.** Source data for *Figure 5*.

the same conditions. Steady state is apparently reached sooner after a voltage pulse in AmH$_v$1 (*Figure 5A*) when compared to hH$_v$1 (*Figure 5C*). The slower kinetics of the human ortholog is also evidenced in the more sluggish deactivation tail currents (*Figure 5C*). The range of voltages over which activation happens is also different between the two channels, with the coral H$_v$1 channel activating at more negative voltages than the human clone (*Figure 5E*; notice that the proton

gradient is such that $\Delta pH = 1$ and is the same for recordings of both channel types). Even though $AmH_v1$ activates at more negative voltages, the activation range is still more positive than the proton reversal potential; thus, coral proton currents activated by depolarization, in the steady state and at least as expressed in HEK293 cells, are always outward.

In order to better estimate both kinetics and activation, we performed experiments with longer pulse durations. This is especially important for the very slow activation of the human channel. The resulting currents are shown in *Figure 5B and D*. The normalized conductance for these currents that are closer to steady state are shown in *Figure 5E* by triangles. These G-V curves are shifted to more negative voltages than the G-V from shorter pulses, as expected. The faster kinetics of $AmH_v1$ is clearly evidenced when the time constant of activation, $\tau$, estimated using fits of the activation time course to *Equation 2*, is compared for coral and human $H_v1$ channels. $AmH_v1$ is more than tenfold faster at 0 mV and over a range of positive voltages (*Figure 5F*).

## Effects of the pH gradient on gating

Both native and cloned voltage-gated proton channels are characteristically modulated by the pH gradient (*Cherny et al., 1995*; *Sasaki et al., 2006*; *Ramsey et al., 2006*). We carried out experiments to investigate the modulation of the coral $H_v1$ channels by different pH gradients. We first recorded whole-cell currents at various $\Delta pH$ values and estimated the voltage-dependence of the conductance. These G-V curves were fitted to *Equation 2* to obtain the voltage of half activation, $V_{0.5}$, and apparent gating charge, $q$, which determines the steepness of the fit. As is the case with other $H_v1$ channels, the $V_{0.5}$ shifts to negative voltages when $\Delta pH$ is greater than 0 and to positive voltages when $\Delta pH$ is less than 0 (*Figure 6A*). When we plot the $V_{0.5}$ as a function of $\Delta pH$, the relationship seems to be mostly linear over the range of $\Delta pH$ −1 to 2. This relationship is somewhat steeper than the generally observed −40 mV/$\Delta pH$ (*Figure 6B*). We tried to obtain recordings over an extended range of $\Delta pH$ values. To this end, we performed inside-out recordings in which the composition of solutions can be better controlled, tends to be more stable, and the size of currents is smaller. However, recordings were unstable at extreme pH values, and we only managed to

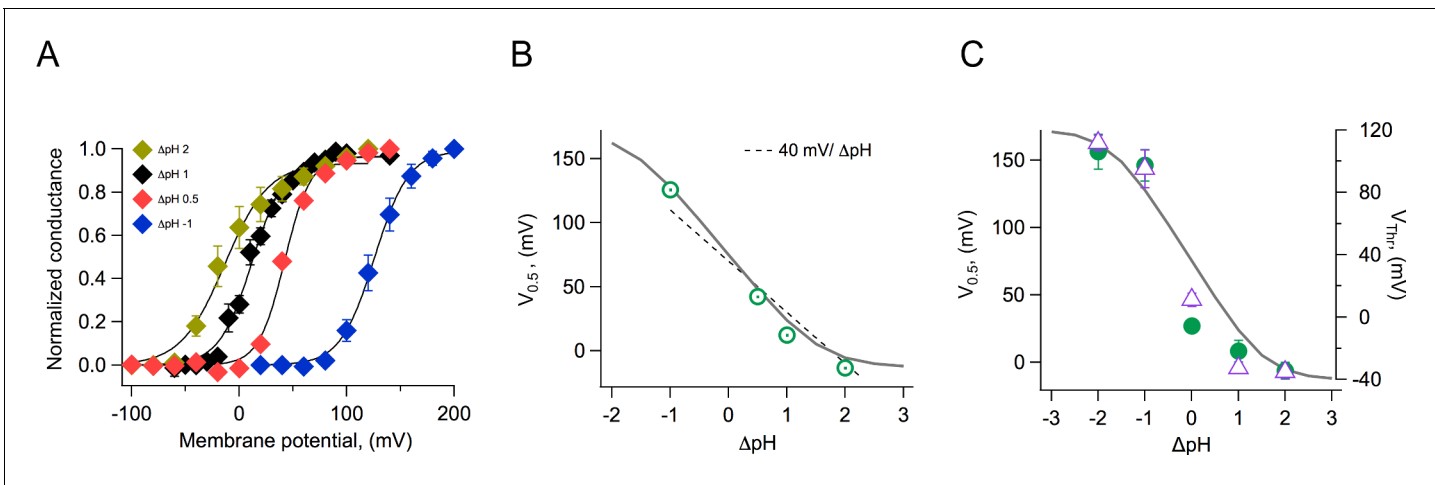

**Figure 6.** Modulation of channel activation by the pH gradient. (**A**) Conductance vs voltage relationships obtained at the indicated $\Delta pH$ values, from whole-cell recordings of *Acropora millepora* $H_v1$ ($AmH_v1$) proton currents. Continuous lines are fits to *Equation 1*. (**B**) The parameter $V_{0.5}$ was obtained from the fits in (**A**) and is displayed as a function of $\Delta pH$. The dotted line is the 40 mV/$\Delta pH$ linear relationship. The continuous gray curve is the prediction of the allosteric model (*Figure 7*). (**C**) Parameters $V_{0.5}$ (green circles) and $V_{Thr}$ (purple triangles) obtained from a different set of inside-out current recordings. Data are mean ± sem. The continuous gray curve is the same prediction of the allosteric model (*Figure 7*) that is shown in panel (**B**). The model parameters used to generate the theoretical curve are E = 5x10⁵, D = 10⁵, C = 0.0002, Kv(0) = 0.00005, $q_g$ = 1.0 $e_o$, $pK_o$ = 3.4, and $pK_i$ = 7. The online version of this article includes the following source data, source code and figure supplement(s) for figure 6:

**Source data 1.** Source data for *Figure 6*.
**Figure supplement 1.** Equations for the model in *Figure 7*.
**Figure supplement 2.** Simulations of the voltage- and pH-dependent behavior predicted by the allosteric model.
**Figure supplement 2—source code 1.** Source code for *Figure 6—figure supplement 2*.

reliably extend the data to a $\Delta$pH value of $-2$. *Figure 6C* shows the summary of the inside-out recordings. We have plotted both the $V_{0.5}$ and the threshold voltage, $V_{Thr}$. To obtain this last parameter, we fitted the exponential rise of the G-V curve to a function of the form

$$G(V) = G' \cdot exp^{qV/K_BT}$$

$V_{Thr}$ was calculated as the voltage at which the fit reaches 10% of the maximum conductance. The parameter $V_{Thr}$ should be less sensitive than $V_{0.5}$ to the possible change in the proton gradient that can occur with large currents. It is clear from these data that at extreme values, the dependence of $V_{0.5}$ or $V_{Thr}$ on $\Delta$pH deviates from a simple linear relationship and instead appears to saturate at extreme values of $\Delta$pH.

## Allosteric model of voltage- and pH-dependent gating

Currently, there is only one quantitative model that has been used to explain $\Delta$pH gating of $H_v1$ channels (*Cherny et al., 1995*). However, this model is euristic and does not provide mechanistic insight into the process of proton modulation of the voltage dependence of proton-permeable channels. In order to explain the modulation of the range of activation by the proton gradient, parameterized by the $V_{0.5}$, we developed a structurally inspired allosteric model of voltage and proton activation. As many voltage-sensing domains, $H_v1$ has two water-occupied cavities exposed to the extracellular and intracellular media (*Ramsey et al., 2010*; *Islas and Sigworth, 2001*; *Ahern and Horn, 2005*). Recent evidence suggests that these cavities function as proton-binding sites through networks of electrostatic interactions (*De La Rosa et al., 2018*). In our model, we propose that these two proton-binding sites, one intracellular and one extracellular, allosterically modulate the movement of the voltage-sensing S4 segment and, thus, channel activation in opposite ways. The extracellular site is postulated as inhibitory, while the intracellular site is excitatory, facilitating voltage sensor movement. As a first approximation, we employed a simplified allosteric formalism based on a Monod-Wyman-Changeux (MWC) style model (*Horrigan and Aldrich, 2002*; *Changeux, 2012*). As a simplifying assumption, in this model we assume that the voltage sensor moves in a single voltage-dependent activation step. We assume the external and internal proton-binding sites have simple protonation given by a single $pK_a$ value. These sites operate as two allosteric modules and are coupled to the voltage sensor according to coupling factors C and D, respectively. These binding sites in turn interact with each other through the coupling factor E. The modular representations of the model are illustrated in *Figure 7A*, while the full model depicting all open and closed states with all permissible transitions and the corresponding equilibrium constants for each transition is shown in *Figure 7B*. Full details of equations derived from these schemes are given in supplementary data.

This allosteric model represents the first attempt at producing a quantitative mechanistic understanding of the interaction of the voltage sensor and protons in $H_v1$ channels.

From the data shown in *Figure 6C*, it can be seen that the model is capable of reproducing the very steep dependence of $V_{0.5}$ on $\Delta$pH and importantly, the saturation of this relationship at extreme values. Some $H_v1$ channels from other organisms show a linear dependence of gating over a large range of $\Delta$pH values, while others show a reduced dependence and even saturation over some range of $\Delta$pH (*Thomas et al., 2018*). Our model can explain these different behaviors as different channels having distinct values of $pK_a$s for the internal or external sites, differences in coupling factors, or differences in the voltage-dependent parameters (*Figure 6—figure supplements 1* and *2*).

## Block by Zn$^{2+}$

The best-characterized blocker of proton channels is the divalent ion zinc (*Cherny et al., 2020*; *De La Rosa et al., 2018*; *Qiu et al., 2016*). We performed experiments to determine if AmH$_v$1 channels are also inhibited by zinc. We found that indeed, extracellular application of zinc in outside-out patches produced inhibition of the channels, reflected in reduced current amplitudes (*Figure 8A*). *Figure 8B* shows average current-voltage (I-V) relationships in the absence and presence of 10 µM external zinc. It can be seen that the fraction of current blocked is not the same at every voltage, indicating that this inhibition might be voltage-dependent. The fraction of blocked channels was calculated and is plotted at each voltage along with the I-V curves (*Figure 8B*). It can be clearly seen that inhibition by Zn$^{2+}$ is voltage-dependent. A simple mechanism for voltage-dependent blockage was proposed by *Woodhull, 1973*. This model postulates that a charged blocker molecule interacts

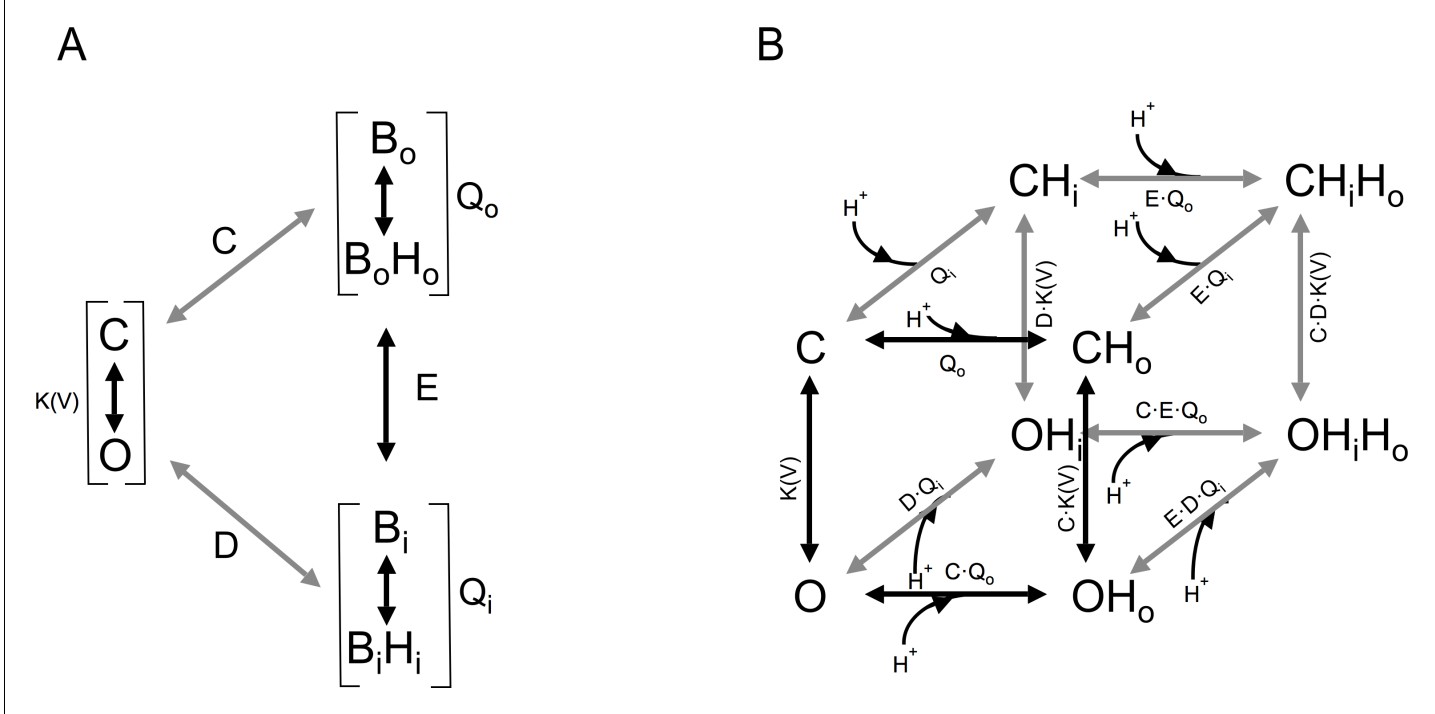

**Figure 7.** Gating scheme I. (**A**) Modular representation of a simple Monod-Wyman-Changeux (MWC) model; the channel opening transition is voltage-dependent, with equilibrium constant $K(V)$. $B_o$ and $B_i$ are the unbound states of the extracellular and intracellular proton-binding sites, respectively, and $B_oH_o$ and $B_iH_i$ are the proton-bound states of these binding sites. $Q_o$ and $Q_i$ are equilibrium constants that depend on the pKa of each of these binding states. C, D, and E are the coupling constants between each of the indicated modules. (**B**) All the individual states implied in (**A**) are depicted, along with proton-binding states and the appropriate equilibrium constants. C, closed states, O, open states. $OH_x$, $OH_xH_x$ and $CH_x$, $CH_xH_x$ are single or double proton-occupied states, where x can be o for outside or i for inside-facing binding sites.

with a binding site in the target molecule that is located within the electric field. Fitting the data according to this model, and given that zinc is a divalent ion, its apparent binding site is located at a fraction $\delta = 0.2$ of the membrane electric field from the extracellular side (*Figure 8B*).

Zinc blockage proceeds very fast. At 1 mM, the channels are blocked almost instantaneously, and the inhibition washes off very fast as well (*Figure 8C*). Finally, we report the dose-response curve (*Figure 8D*). The inhibition dose-response curve can be fit by a Hill equation (*Equation 4*) with a slope factor of nearly 0.5 and an apparent dissociation constant, $K_D$, of 27 µM.

## Discussion

A few ion transport mechanisms in reef-building corals have been described, but up to now, no ion channels have been characterized from any scleractinian species. Here we have shown that voltage-gated proton-permeable channels formed by the $H_v1$ protein are present in corals. In particular, we have cloned these channels from two species of the genus *Acropora*, *A. millepora* and *A. palmata.* It is interesting that the protein sequence of these proteins shows a very high degree of conservation, suggesting that, even when the two species are found in different oceans, they haven't had time to diverge substantially or alternatively; selective pressures on these channels are very similar in both species. The presence of $H_v1$ sequences in many other species of corals from disparate clades suggests that $H_v1$ plays an important role in coral physiology.

$H_v1$ channels are formed by a protein fold that is structurally equivalent to the VSDs of canonical voltage-gated channels (*Sasaki et al., 2006*; *Ramsey et al., 2006*). The VSD is formed by a bundle of four antiparallel alpha helices (*Takeshita et al., 2014*). In some species, it has been shown that $H_v1$ channels are dimeric (*Lee et al., 2008*; *Mony et al., 2020*; *Lee et al., 2008*). Accordingly, we have also shown here that the $AmH_v1$ is a dimer. Our FRET results are consistent with the high propensity to form a coiled coil shown by its C-terminal domain.

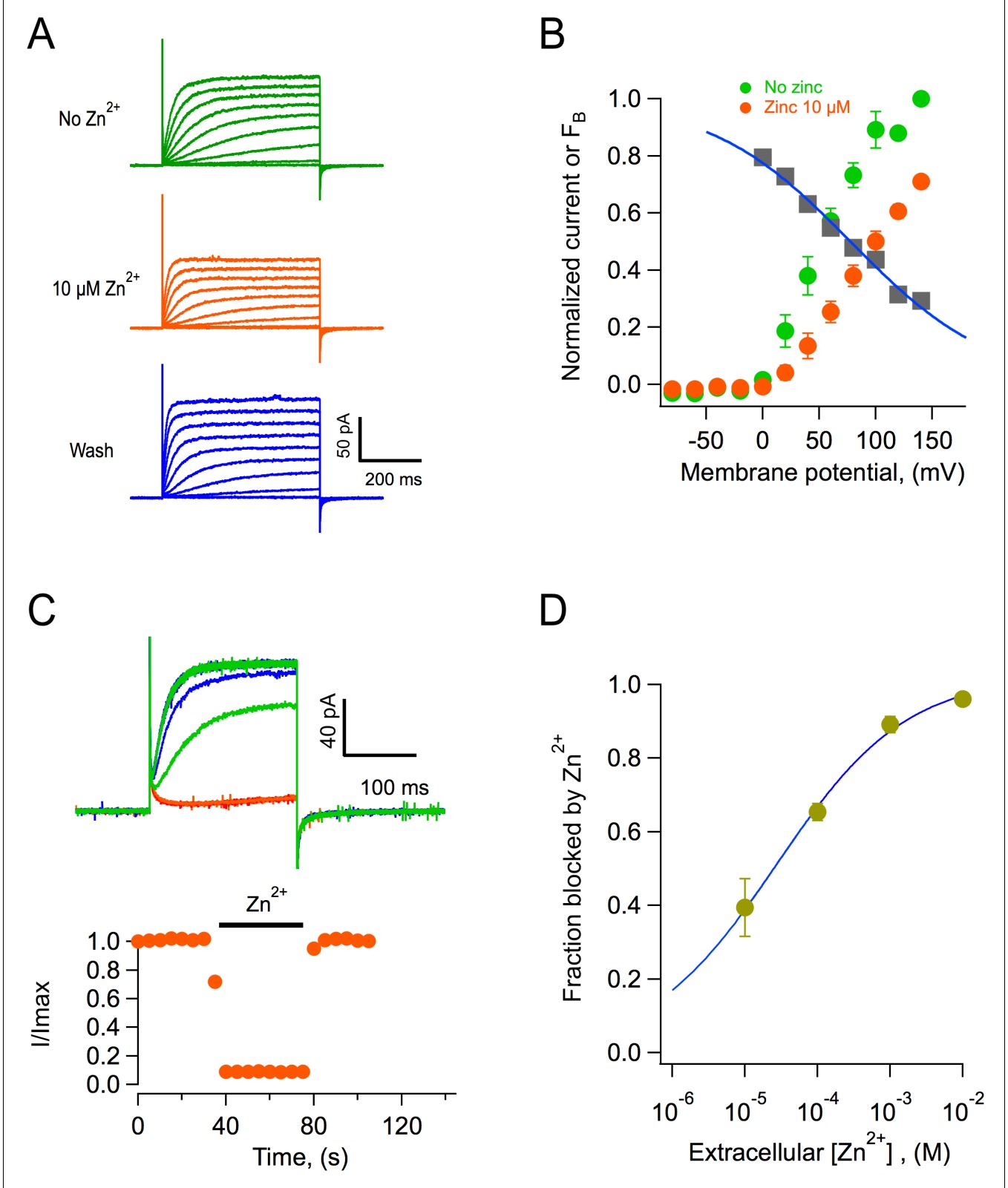

**Figure 8.** Block of AmH$_v$1 channels by extracellular zinc. (**A**) *Acropora millepora* H$_v$1 (AmH$_v$1)-mediated currents from an outside-out patch in the absence (top) and presence of 10 µM zinc (middle) and after washing of zinc (bottom). The scale bars apply to the three current families. Voltage pulses lasted 500 ms and were from −80 to 140 mV in 20 mV steps. The ΔpH = 1 with pH$_i$ = 6 and pH$_o$ = 7. (**B**) Normalized current-voltage relationships before and in the presence of 10 µM zinc from four patches as in (**A**). The gray squares are the ratio I$_{zinc}$(V)/I(V), which gives the voltage-dependence of

*Figure 8 continued on next page*

*Figure 8 continued*

the blocking reaction. The blue curve is the fit to the Woodhull equation $F_B = \frac{1}{1+e^{\frac{-\delta z(V-V_{0.5})}{K_B T}}}$ , where $F_B$ is the fraction of the current blocked, $\delta$ is the

fraction of the electric field where the blocker binds, $z$ is the valence of the blocker, $V_{0.5}$ is the potential where half of the current is blocked, $K_B$ is Boltzmann's constant, and T is the temperature in Kelvin. The fitting parameters are $\delta = 0.19$, $V_{0.5} = 77.6$ mV. (C) The effect of zinc is fast. Application of 1 mM zinc to an outside-out patch produces almost instantaneous block of ~90% of the current. The effect also washes off quickly upon removal of zinc. Trace colors are as in (A). Voltage pulse was 100 mV applied every 5 s. (D) Dose-response curve of zinc block of AmH$_v$1 obtained at 100 mV. The continuous curve is a fit of the data to *Equation 4* with apparent $K_D = 27.4$ μM and n = 0.48.

The online version of this article includes the following source data for figure 8:

**Source data 1.** Source data for *Figure 8*.

H$_v$1 channels are different from canonical voltage-gated channels in that both voltage sensing and permeation are mediated through a single protein domain. Voltage sensing is thought to occur through the interaction of charged amino acid side chains with the electric field, leading to the outward movement of the fourth domain or S4, in a similar fashion to other voltage-sensing domains (*Carmona et al., 2018*; *De La Rosa and Ramsey, 2018*). This outward movement of the S4 is coupled to protons moving through the VSD in a manner that is not completely understood (*Randolph et al., 2016*). Most proton-permeable channels seem to have evolved to extrude protons from the cell, and toward this end, their voltage dependence is tightly modulated by the proton gradient between extracellular and intracellular solutions (*Cherny et al., 1995*).

Our electrophysiology experiments show that these coral channels give rise to proton currents when expressed in HEK293 cells and that they retain the functional characteristics that have been shown to define the class in other species, such as very high selectivity for protons, activation by voltage, and modulation of this activation by the proton gradient. The new channels reported here activate faster than the human H$_v$1 channel. It has been known that different orthologs of H$_v$1 activate with varying kinetics. For example, sea urchin, dinoflagellate, and recently, fungal H$_v$1 channels activate rapidly, while most mammalian counterparts have slow activation rates (*Musset et al., 2008*; *Smith et al., 2011*; *Zhao and Tombola, 2021*). A comparative study suggests that two amino acids in the S3 transmembrane segment are important determinants of kinetic differences between sea urchin and mouse H$_v$1 (*Sakata et al., 2016*). The authors suggest that the time course of activation is slow in channels containing a histidine and a phenylalanine at positions 164 and 166, respectively (mouse sequence numbering). The AmH$_v$1 has a histidine at an equivalent position, 132, and a methionine at 134. It is possible that this last amino acid in AmH$_v$1 confers most of the fast kinetics phenotype. A separate work has shown that a lack of the amino-terminal segment in human sperm H$_v$1 also produced fast-activating channels (*Berger et al., 2017*). Interestingly, the *Acropora* channels have a shorter amino-terminal sequence, which could also contribute to their fast kinetics.

One of the most interesting characteristics found in these new proton channels is their modulation by the proton gradient. As opposed to other H$_v$1 channels, we can observe a trend toward saturation of the V$_{0.5}$ for activation as a function of ΔpH at extreme values of this variable. A tendency toward saturation of the V$_{0.5}$-ΔpH relationship has been observed in mutants of the hH$_v$1 channel (*Cherny et al., 2015*) or at negative values of ΔpH for a snail H$_v$1 (*Thomas et al., 2018*), but it seems it can be fully appreciated in AmH$_v$1. Since our model explains the observation of saturation of voltage gating at extreme values of ΔpH as a consequence of the existence of two saturable sites for proton binding, we attribute this behavior to the large separation of pK$_a$ values for the extracellular and intracellular proton-binding sites. Thus, channels that do not show saturation might have sites with well-separated and differing pKa values. Evolutionary fine tuning of these pKa values will produce channels with pH activation in ranges adapted to their physiological function.

It is important to point out that the external and internal proton-binding sites can be part of the voltage sensor itself. We envision these sites as being formed not by distinct protonatable amino acids but by a network of electrostatic interactions between amino acids in S4 and other transmembrane segments. In this context, the pK$_a$ values of the sites in our model do not reflect pK$_a$s of individual amino acids but of the whole proton-binding site. Recent works by *Carmona et al., 2021* and *Schladt and Berger, 2020* suggest that voltage sensing is directly responsible for pH gating. These authors suggest that the energy of the pH gradient is part of the free energy needed for

voltage-sensor movement. This purely energetic statement is encompassed in our model as the coupling factors between voltage-sensor movement and proton binding, which are in fact energetic factors.

In our model, the strength of allosteric coupling of these sites and the voltage sensor determines if saturation is observed over a short or extended range of $\Delta pH$ values and the range of values of $V_{0.5}$ that a particular channel can visit. Coupling also determines the value of the slope of the $\Delta pH$ vs $V_{0.5}$ curve. Values larger or smaller than the typical ~40 mV/pH are obtained as a consequence of strong or weak coupling between binding and voltage-sensor movement, respectively (*Figure 6— figure supplement 2*). Our model should provide a framework to better understand the gating mechanisms in future work.

It is clear that more complicated models, with a larger number of voltage-dependent and -independent steps (*Villalba-Galea, 2014*) and coupling to protonation sites, should be the next step to improve data fitting and explore voltage- and proton-dependent kinetics. In particular, these types of models can help explain mutagenesis experiments exploring the nature of the protonation sites. For example, *Villalba-Galea, 2014* proposed the existence of a voltage-independent step in Hv1 gating, and this can be easily incorporated in the model proposed here, as is the fact that $hH_v1$ and $AmH_v1$ channels are dimers. The present form of the allosteric model has a single subunit. Since gating in dimeric channels seems to be cooperative, allosteric models have the advantage that cooperativity between subunits can be handled naturally by including coupling between two voltage sensors.

$H_v1$ proton channels seem fundamental in handling fluctuations in intracellular pH and take part in several well-characterized physiological processes that depend on proton concentration changes, such as intracellular pH regulation, sperm flagellum beating, reactive oxygen species production and bacterial killing in immune cells, initiation of bioluminescence in single-celled algae, and so on *Castillo et al., 2015*.

What is the function of voltage-gated proton channels in corals? The deposition of a $CaCO_3$ exoskeleton is one of the main defining characteristics of scleractinians; however, the ionic transport mechanisms involved in this process are mostly unknown. In order for aragonite precipitation to occur favorably, the pH of the calicoblastic fluid, right next to the skeleton, is maintained at high levels, between 8.5 and 9 and above the pH of sea water (*Le Goff et al., 2017*). It has been posited that corals control this pH via vectorial transport of protons to the gastrodermal cavity (*Jokiel, 2013*). Since proton transport away from the site of calcification would incur a drastically lower intracellular pH in the cells of the aboral region, we propose that, given their ability to rapidly regulate the intracellular pH (*De-la-Rosa et al., 2016*), $H_v1$ proton channels contribute by transporting protons from the cells. Thus, these proton channels could be a major component of the mechanisms of intracellular pH regulation in corals. Given that the activation range of $H_v1$ is controlled by the pH gradient, a large intracellular acidification would facilitate opening of these channels at the resting potential of cells, which is presumably negative.

The finding that coral $H_v1$ channels retain their sensitivity to $Zn^{2+}$ opens the possibility of using this ion as a pharmacological tool to study the role of proton channels in pH homeostasis. It is interesting that a recent report has shown detrimental effects of zinc supplementation on coral growth (*Tijssen et al., 2017*), a result that could be explained by zinc inhibition of $H_v1$.

The physiological role of $H_v1$ channels in corals might be essential in the response of these organisms to ocean acidification. Although further research is needed to understand the cellular and subcellular localization of these channels, we theorize that as the pH of sea water acidifies, gating of $H_v1$ should require stronger depolarization, thus hindering its capacity to transport protons from the cell. This will contribute to a diminished calcification rate and less aragonite saturation of the $CaCO_3$ skeleton. It would be interesting and important to study the effects of acidification on $H_v1$ physiology and pH regulation in corals in vivo. Essentially nothing is known about the electrophysiological properties of coral cells. This report represents the first time that an ion channel has been cloned and characterized in any coral and should open a new avenue of research, such as uncovering the cellular and possible subcellular localization of these channels and carefully measuring their physiological role in vivo.

# Materials and methods

## Identification of H$_v$1 sequences and cloning

Blast searches of the transcriptome of the Indo-Pacific coral *A. millepora* (*Moya et al., 2012*) detected four sequences that we identified as belonging to a putative proton-permeable channel. The GenBank accession numbers for these are XM_015907823.1, XM_015907824.1, XM_029346499.1, and XM_029346498.1. We designed two pairs of oligonucleotides to amplify two of these sequences (*Table 1*). Total RNA was extracted from the tissue obtained from a fragment of *A. millepora* acquired from a local salt-water aquarium provider (Reefservices, Mexico City). RNA was extracted by dipping the whole fragment for 2 min in 5 ml of solution D (4 M guanidinium thio-cyanate, 25 mM sodium citrate, 5% sarkosyl, and 0.1 M 2-mercaptoethanol). After incubation, the tissue was removed by gently pipetting the solution for 2 min. At this point, the calcareous skele-ton was removed and RNA extraction continued according to *Chomczynski and Sacchi, 1987*. Total RNA (1 µg) from *A. millepora* was used for reverse transcription polymerase chain reaction (RT-PCR), employing oligo dT and SuperScripII reverse transcriptase (Invitrogen, USA). cDNA obtained from RT-PCR was used in three PCRs using oligos (1) AcHv1Nter5´ and 3´, (2) AcHv1Cter5´ and 3´, and (3) AcHv1Nter5´ and AcHv1Cter3´ (*Table 1*). The Platinium Pfx DNA polymerase (Invitrogen) was used for amplification according to the manufacturer's instructions. 1 µl of Taq DNA polymerase (Invitro-gen, USA) was used for 10 min at 72°C to add a poly-A tail at 5' and 3' ends and facilitate cloning into the pGEM-T vector.

The PCR 3 gave rise to a full open reading frame (ORF) containing AmH$_v$1. New oligos, AcHv1Nter5´ and AcHv1Cter3´, containing restriction sites Kpn1 and Not1, respectively, were used to re-amplify the ORF in pGEM-T and subclone it into pcDNA3.1 for heterologous expression.

The H$_v$1 channel from *A. palmata* was cloned from a fragment of an adult specimen collected from the Limones Reef off of Puerto Morelos, Mexico. RNA extraction from small coral pieces was carried out by flash freezing in liquid nitrogen and grinding the frozen tissue. All other cloning proce-dures were as for *A. millepora*. All clones and DNA constructs were confirmed by automatic sequencing at the Molecular Biology Facility of the Instituto de Fisiología Celular at UNAM.

## Heterologous expression of AmH$_v$1

All electrophysiological and FRET experiments were carried out in HEK293 cells heterologously expressing the specified clone or DNA construct. HEK293 cells were acquired from ATCC (Gaithers-burg, MD, USA) and were found to be free of mycoplasma infection using a PCR-based detection kit (Sigma-Aldrich, USA). Cells were grown on 100 mm culture dishes with 10 ml of Dulbecco's Modified Eagle Medium (DMEM, Invitrogen) containing 10% fetal bovine serum (FBS) (Invitrogen, USA) and 100 units/ml-100 µg/ml of penicillin-streptomycin (Invitrogen, USA), incubated at 37°C in an incuba-tor with 5.2% $CO_2$ atmosphere. When cells reached 90% confluence, the medium was removed, and the cells were treated with 1 ml of 0.05% trypsin-ethylenediaminetetraacetic acid (EDTA) (Invitrogen, USA) for 5 min. Subsequently, 1 ml of DMEM with 10% FBS was added. The cells were mechanically dislodged and reseeded in 35 mm culture dishes over 5x5 mm coverslips for electrophysiology or in 35 mm glass bottom dishes for FRET experiments. In both cases, 2 ml of complete medium was used. Cells at 70% confluence were transfected with pcDNA3.1-AmH$_v$1 prepared from a plasmid midiprep, using jetPEI transfection reagent (Polyplus Transfection, France). For patch-clamp experi-ments, pEGFP-N1 (BD Biosciences Clontech, USA) was cotransfected with the channel DNA to

**Table 1.** Oligonucleotides used to clone amino- and carboxy-terminal partial sequences of AmH$_v$1 from total reverse-transcribed mRNA from *A. millepora*.

| Oligo name | Sequence |
| --- | --- |
| AcHvNt5´ | ATGATTGATGCAAGAACCAGACGATCGAGCATGGATGAT |
| AcHvNt3´ | TGATCCTGCTCTCAAGTCAAGAACCAACTCAGCAATGAC |
| AcHvCt5´ | ATGGGATTCACATTTTCAAGCACAAATGGAGGTGTTT |
| AcHvCt3´ | TCAGCTTTGTTTTAATGTTGTCAATTCAGACTCCAACTG |

visualize successfully transfected cells via their green fluorescence. Electrophysiological recordings were done 1 or 2 days after transfection.

## FRET measurement of stoichiometry

In order to measure the stoichiometry of subunit interaction employing FRET, we constructed fusion proteins between $AmH_v1$ and mCerulean and mCitrine FPs, to be used as the donor and acceptor, respectively. The FPs were fused to the N-terminus of the channel in order to disrupt as little as possible the C-terminus-mediated interaction. These constructs were transfected into HEK293 cells as described above. The apparent FRET efficiency between FP-containing constructs, $E_{app}$, was measured via sensitized emission of the acceptor, employing the spectral-FRET method (*De-la-Rosa et al., 2013*; *Zheng et al., 2002*). Fluorescence was measured in a home-modified TE-2000U inverted epifluorescense microscope (Nikon, Japan). The excitation light source was an Argon Ion laser (Spectra-Physics, Germany) mainly producing light at 458, 488, and 514 nm; the laser beam was focused and then collimated using a 3 mm ball lens and a 50-mm focal length planoconvex lens. Collimated light is steered with a mirror and then is focused into the objective back focal plane by a 300-mm focal length achromatic lens.

Cells were imaged with a Nikon 60x oil immersion objective (numerical aperture 1.4). The detection arm of the microscope is coupled to a spectrograph (Acton Instruments, USA) and an EM-CCD camera (Ixon Ultra, Andor, Ireland) controlled by Micromanager software (*Edelstein et al., 2014*). Excitation was achieved with appropriate excitation filters (Chroma, Vermont, USA) for mCerulean (458 nm) and mCitrine (488 nm). The emission filter was a long-pass filter in order to collect the full emission spectrum of the FRET pair. Apparent FRET efficiency is plotted as a function of the fluorescence intensity ratio ($I_{donor}/I_{acceptor}$). This relationship can be fitted with models of subunit association with a fixed stoichiometry, according to *De-la-Rosa et al., 2013*.

## Electrophysiology

All chemicals for solutions were acquired from Sigma-Aldrich (Mexico). Proton current recordings were made from HEK293 cells expressing pCDNA3.1-$AmH_v1$ in the inside-out, whole-cell, and outside-out configurations of the patch-clamp recording technique. For whole-cell and inside-out recordings, the extracellular solution (bath and pipette, respectively) was (in mM) 80 tetramethylammonium and methanesulfonic acid ($TMA-HMESO_3$), 100 buffer ((2-(N-morpholino)ethanesulfonic acid (MES): pH 5.5, 6.0, and 6.5; 4-(2-hydroxyethyl)-1-piperazineethanesulfonic acid (HEPES): pH 7.0, 7.5), 2 $CaCl_2$, 2 $MgCl_2$, and pH-adjusted N-methyl-d-glucamine/tetramethylammonium hydroxide (NMDG/TMA-OH) and HCl. The intracellular solution (pipette and bath, respectively) was (in mM) 80 $TMA-HMESO_3$, 100 buffer (MES: pH 5.5, 6.0, and 6.5; HEPES: pH 7.0, 7.5), 1 ethylene glycol tetraacetic acid (EGTA), and pH-adjusted NMDG/TMA-OH and HCl.

Macroscopic currents were low-pass filtered at 2.5 kHz, sampled at 20 kHz with an Axopatch 200B amplifier (Axon Instruments, USA) using an Instrutech 1800 AD/DA board (HEKA Elektronik, Germany) or an EPC-10 amplifier (HEKA Elektronik, Germany). Acquisition control and initial analysis were done with PatchMaster software. Pipettes for recording were pulled from borosilicate glass capillaries (Sutter Instrument, USA) and fire-polished to a resistance of 4–7 MΩ when filled with recording solution for inside- and outside-out recordings and 1–3 MΩ for the whole-cell recording. The bath (intracellular) solutions in inside-out patches were changed using a custom-built rapid solution changer. For whole-cell recordings, all the bath solution was exchanged to manipulate pH. In some recordings, linear current components were subtracted using a p/4 subtraction protocol.

## Conditions for recording zinc effects

The effect of zinc was evaluated in outside-out patches at a ΔpH of 1. The bath solution composition was (in mM) 100 $TMA-HMESO_3$, 100 HEPES, 8 HCl, 2 $CaCl_2$, 2 $MgCl_2$, and the indicated concentration of $ZnCl_2$. The pipette solution was (in mM) 100 $TMA-MESO_3$, 100 MES, 8 HCl, 10 EGTA, and 2 $MgCl_2$. Both solutions were adjusted to pH 7 and pH 6, respectively, with TMA-OH/HCl. Patches were placed in front of a perfusion tube that was gravity-fed with the appropriate solution. Tubes were changed with a home-built rapid perfusion system.

## Data analysis

Conductance, G, was calculated from I-V relations assuming ohmic instantaneous currents, according to

$$I(V) = G \cdot (V - V_{rev})$$

The normalized G-V relations were fit to a Boltzmann function according to *Equation 1,*

$$\frac{G}{G_{max}} = \frac{1}{1 + \exp\left(\frac{q(V - V_{0.5})}{K_B T}\right)} \tag{1}$$

Here, $V_{0.5}$ is the voltage at which $G/G_{max} = 0.5$, $q$ is the apparent gating charge (in elementary charges, $e_o$), $K_B$ is the Boltzmann constant, and $T$ is the temperature in Kelvin (22°C).

The time constant of activation was estimated via a fit of the second half of currents to the equation

$$I(t) = I_{ss} \cdot \left(1 - e^{\left(\frac{-(t-\delta)}{\tau}\right)}\right) \tag{2}$$

where $I_{ss}$ is the amplitude of the current at steady state, $\delta$ is the delay of the exponential with respect to the start of the voltage pulse, and $\tau$ is the time constant, both with units of ms. The voltage-dependence of $\delta$ and $\tau$ was estimated from a fit to equation

$$k(V) = k(0)e^{(-Vq_i/K_B T)} \tag{3}$$

where $i$ stands for $\delta$ or $\tau$ and $k(0)$ is the value of either parameter at 0 mV.

Currents in the presence of zinc were normalized to the current before application of the ion to obtain a normalized fraction of current blocked as $F_B = 1 - I/I_{max}$. The zinc dose-response curve was fitted to Hill's equation in the form

$$F_B = \frac{1}{1 + \left(\frac{K_D}{[Zn^{2+}]_o}\right)^{n_H}} \tag{4}$$

where $K_D$ is the apparent dissociation constant, $[Zn^{2+}]_o$ is the extracellular zinc concentration, and $n_H$ is the Hill coefficient.

## Acknowledgements

We would like to thank Itzel A Llorente for the excellent technical assistance. We also thank Gerardo Coello from IFC-UNAM for help with analysis of coral transcriptomes. This work was funded in part by grant no. IN215621 from DGAPA-PAPIIT-UNAM to LDI, grant no. 247765 to ATB, and grant no. IN200720 to TR. EM was funded by Conacyt-Fronteras en la Ciencia Grant No. 2.

## Additional information

### Competing interests

Leon D Islas: Reviewing editor, *eLife*. The other authors declare that no competing interests exist.

### Funding

| Funder | Grant reference number | Author |
|---|---|---|
| Universidad Nacional Autónoma de México | IN215621 | Leon D Islas |
| Universidad Nacional Autónoma de México | IN200720 | Tamara Rosenbaum |
| Universidad Nacional Autónoma de México | IN247765 | Anastazia T Banaszak |

| Consejo Nacional de Ciencia y Tecnología | 2 | Ernesto Maldonado |

The funders had no role in study design, data collection and interpretation, or the decision to submit the work for publication.

## Author contributions

Gisela Rangel-Yescas, Resources, Investigation, Project administration, Writing - review and editing; Cecilia Cervantes, Miguel A Cervantes-Rocha, Esteban Suárez-Delgado, Formal analysis, Investigation, Methodology, Writing - review and editing; Anastazia T Banaszak, Resources, Funding acquisition, Writing - review and editing; Ernesto Maldonado, Resources, Funding acquisition, Methodology, Writing - review and editing; Ian Scott Ramsey, Conceptualization, Writing - review and editing; Tamara Rosenbaum, Resources, Software, Formal analysis, Validation, Visualization, Writing - original draft, Writing - review and editing; Leon D Islas, Conceptualization, Resources, Data curation, Software, Formal analysis, Supervision, Funding acquisition, Validation, Investigation, Visualization, Methodology, Writing - original draft, Project administration, Writing - review and editing

## Author ORCIDs

Miguel A Cervantes-Rocha [iD] http://orcid.org/0000-0001-7401-7567
Esteban Suárez-Delgado [iD] http://orcid.org/0000-0003-0147-3451
Anastazia T Banaszak [iD] http://orcid.org/0000-0002-6667-3983
Ernesto Maldonado [iD] http://orcid.org/0000-0002-9627-967X
Ian Scott Ramsey [iD] http://orcid.org/0000-0002-6432-4253
Leon D Islas [iD] https://orcid.org/0000-0002-7461-5214

## Decision letter and Author response

Decision letter https://doi.org/10.7554/eLife.69248.sa1
Author response https://doi.org/10.7554/eLife.69248.sa2

# Additional files

## Supplementary files

- Transparent reporting form

## Data availability

All data generated or analyzed during this study are included in the manuscript and supporting files. We have provided an Excel file with source data used for figures.

The following datasets were generated:

| Author(s) | Year | Dataset title | Dataset URL | Database and Identifier |
|---|---|---|---|---|
| Rangel-Yescas G, Cervantes C, Cervantes-Rocha MA, Suárez-Delgado E, Banaszak AT, Maldonado E, Ramsey IS, Rosenbaum T, Islas LD | 2021 | Acropora millepora Hv1 proton-channel nucleotide sequence. MZ029047 - Acropora palmata Hv1 proton-channel nucleotide sequence. | https://www.ncbi.nlm.nih.gov/nuccore/MZ029046 | NCBI GenBank, MZ029046 |
| Rangel-Yescas G, Cervantes C, Cervantes-Rocha MA, Suárez-Delgado E, Banaszak AT, Maldonado E, | 2021 | Acropora palmata Hv1 proton-channel nucleotide sequence | https://www.ncbi.nlm.nih.gov/nuccore/MZ029047 | NCBI GenBank, MZ029047 |

Ramsey IS,
Rosenbaum T, Islas
LD

The following previously published datasets were used:

| Author(s) | Year | Dataset title | Dataset URL | Database and Identifier |
|---|---|---|---|---|
| Australian National University | 2019 | Title voltage-gated hydrogen channel 1-like [Acropora millepora] | https://www.ncbi.nlm.nih.gov/protein/1666377134/ | NCBI Protein, XP_029202331.1 |
| King Abdullah University of Science and Technology | 2017 | voltage-gated hydrogen channel 1-like [Stylophora pistillata] | https://www.ncbi.nlm.nih.gov/protein/XP_022795192.1 | NCBI Protein, XP_022795192.1 |
| University of Miami | 2018 | voltage-gated hydrogen channel 1-like [Pocillopora damicornis] | https://www.ncbi.nlm.nih.gov/protein/XP_027057117.1 | NCBI Protein, XP_027057117.1 |
| QUT | 2019 | voltage-gated hydrogen channel 1-like [Actinia tenebrosa] | https://www.ncbi.nlm.nih.gov/protein/XP_031564162.1 | NCBI Protein, XP_031564162.1 |
| Joint Genome Institute (JGI) | 2017 | voltage-gated hydrogen channel 1 [Nematostella vectensis] | https://www.ncbi.nlm.nih.gov/protein/XP_001626501.1 | NCBI Protein, XP_001626501.1 |
| JCVI | 2009 | PREDICTED: voltage-gated hydrogen channel 1-like [Hydra vulgaris] | https://www.ncbi.nlm.nih.gov/protein/828190613/ | NCBI Protein, XP_012554112.1 |
| Anon | 2012 | voltage-gated proton channel [Strongylocentrotus purpuratus] | https://www.ncbi.nlm.nih.gov/protein/187282419/ | NCBI Protein, NP_001119779.1 |
| Anon | 2006 | voltage-gated hydrogen channel 1 [Ciona intestinalis] | https://www.ncbi.nlm.nih.gov/protein/NP_001071937.1 | NCBI Protein, NP_001071937.1 |
| Vertebrate Genomes Project | 2020 | voltage-gated hydrogen channel 1 [Petromyzon marinus] | https://www.ncbi.nlm.nih.gov/protein/XP_032803138.1 | NCBI Protein, XP_032803138.1 |
| Wellcome Sanger Institute | 2021 | voltage-gated hydrogen channel 1 [Rana temporaria] | https://www.ncbi.nlm.nih.gov/protein/XP_040202566.1 | NCBI Protein, XP_040202566.1 |
| Naturalis Biodiversity Center | 2013 | Voltage-gated hydrogen channel 1, partial [Ophiophagus hannah] | https://www.ncbi.nlm.nih.gov/protein/565320699/ | NCBI Protein, ETE71598.1 |
| Baylor College of Medicine | 2005 | voltage-gated hydrogen channel 1-like isoform X1 [Strongylocentrotus purpuratus] | https://www.ncbi.nlm.nih.gov/protein/XP_030847861.1 | NCBI Protein, XP_030847861.1 |
| China Agricultural University | 2010 | Voltage-gated hydrogen channel 1, partial [Anas platyrhynchos] | https://www.ncbi.nlm.nih.gov/protein/EOA95241.1 | NCBI Protein, EOA95241.1 |
| Anon | 2002 | voltage-gated hydrogen channel 1 [Xenopus laevis] | https://www.ncbi.nlm.nih.gov/protein/NP_001088875.1 | NCBI Protein, NP_001088875.1 |
| KAUST | 2015 | Voltage-gated hydrogen channel 1 [Exaiptasia diaphana] | https://www.ncbi.nlm.nih.gov/protein/KXJ27230.1 | NCBI Protein, KXJ27230.1 |

| | | | | |
|---|---|---|---|---|
| The University of Queensland | 2020 | voltage-gated hydrogen channel 1 [Cygnus atratus] | https://www.ncbi.nlm.nih.gov/protein/XP_035421542.1 | NCBI Protein, XP_035421542.1 |
| Anon | 2005 | voltage-gated hydrogen channel 1 [Gallus gallus] | https://www.ncbi.nlm.nih.gov/protein/NP_001025834.1 | NCBI Protein, NP_001025834.1 |
| Vertebrate Genomes Project | 2019 | voltage-gated hydrogen channel 1 [Ornithorhynchus anatinus] | https://www.ncbi.nlm.nih.gov/protein/XP_028914661.1 | NCBI Protein, XP_028914661.1 |
| University of Washington | 2018 | voltage-gated hydrogen channel 1 [Theropithecus gelada] | https://www.ncbi.nlm.nih.gov/protein/XP_025257726.1 | NCBI Protein, XP_025257726.1 |
| Anon | 2002 | HVCN1 protein [Homo sapiens] | https://www.ncbi.nlm.nih.gov/protein/AAH07277.1 | NCBI Protein, AAH07277.1 |
| The Roslin Institute | 2020 | voltage-gated hydrogen channel 1 isoform X1 [Canis lupus familiaris] | https://www.ncbi.nlm.nih.gov/protein/XP_038292573.1 | NCBI Protein, XP_038292573.1 |
| Anon | 2021 | voltage-gated hydrogen channel 1 [Mus musculus] | https://www.ncbi.nlm.nih.gov/protein/NP_001035954.1 | NCBI Protein, NP_001035954.1 |
| Beijing Genomics Institute | 2003 | voltage-gated hydrogen channel 1 [Mus musculus] | https://www.ncbi.nlm.nih.gov/protein/XP_005424087.1 | NCBI Protein, XP_005424087.1 |
| College of Medicine and Forensics, Xi'an Jiaotong University | 2014 | voltage-gated hydrogen channel 1 [Egretta garzetta] | https://www.ncbi.nlm.nih.gov/protein/XP_009633183.1 | NCBI Protein, XP_009633183.1 |
| Vertebrate Genomes Project | 2020 | voltage-gated hydrogen channel 1 [Amblyraja radiata] | https://www.ncbi.nlm.nih.gov/protein/XP_032899181.1 | NCBI Protein, XP_032899181.1 |
| US Department of Agriculture, Agriculture Research Service | 2018 | voltage-gated hydrogen channel 1 [Fusarium longipes] | https://www.ncbi.nlm.nih.gov/protein/RGP61076.1 | NCBI Protein, RGP61076.1 |
| US Department of Agriculture, Agriculture Research Service | 2018 | voltage-gated hydrogen channel 1 [Fusarium flagelliforme] | https://www.ncbi.nlm.nih.gov/protein/RFN53390.1 | NCBI Protein, RFN53390.1 |
| US Department of Agriculture, Agriculture Research Service | 2020 | voltage-gated hydrogen channel 1 [Fusarium heterosporum] | https://www.ncbi.nlm.nih.gov/protein/KAF5660113.1 | NCBI Protein, KAF5660113.1 |
| The Institute of Vegetables and Flowers CAAS | 2019 | voltage-gated hydrogen channel 1 [Cordyceps javanica] | https://www.ncbi.nlm.nih.gov/protein/TQW00298.1 | NCBI Protein, TQW00298.1 |
| Broad Institute | 2006 | voltage-gated hydrogen channel 1 [Aplysia californica] | https://www.ncbi.nlm.nih.gov/protein/XP_005100666.1 | NCBI Protein, XP_005100666.1 |
| The Genomic Institute | 2017 | Voltage-gated hydrogen channel 1 [Fasciola hepatica] | https://www.ncbi.nlm.nih.gov/protein/THD25470.1 | NCBI Protein, THD25470.1 |
| Global Invertebrate Genomics Alliance (GIGA) | 2020 | HVCN1 [Bugula neritina] | https://www.ncbi.nlm.nih.gov/protein/KAF6036357.1 | NCBI Protein, KAF6036357.1 |
| NINGBO UNIVERSITY | 2020 | HVCN1 [Mytilus coruscus] | https://www.ncbi.nlm.nih.gov/protein/CAC5426376.1 | NCBI Protein, CAC5426376.1 |
| McDonnell Genome Institute | 2020 | Voltage-gated hydrogen channel 1 [Paragonimus heterotremus] | https://www.ncbi.nlm.nih.gov/protein/KAF5400532.1 | NCBI Protein, KAF5400532.1 |

| | | | | |
|---|---|---|---|---|
| Fudan University | 2019 | Voltage-gated hydrogen channel 1 [Schistosoma japonicum] | https://www.ncbi.nlm.nih.gov/protein/TNN21174.1 | NCBI Protein, TNN21174.1 |
| BGI-Shenzhen | 2012 | voltage-gated hydrogen channel 1 [Cicer arietinum] | https://www.ncbi.nlm.nih.gov/protein/XP_012568882.1 | NCBI Protein, XP_012568882.1 |
| The Cucumber Genome Initiative | 2009 | voltage-gated hydrogen channel 1 [Cucumis sativus] | https://www.ncbi.nlm.nih.gov/protein/XP_011656484.2 | NCBI Protein, XP_011656484.2 |
| African Centre of excellence in Phytomedicine Resaerch | 2018 | Voltage-gated hydrogen channel 1, partial [Mucuna pruriens] | https://www.ncbi.nlm.nih.gov/protein/RDX63547.1 | NCBI Protein, RDX63547.1 |
| Xi'an Jiaotong University | 2018 | voltage-gated hydrogen channel 1 [Papaver somniferum] | https://www.ncbi.nlm.nih.gov/protein/XP_026460796.1 | NCBI Protein, XP_026460796.1 |
| International peanut genome project | 2018 | Voltage-gated hydrogen channel [Arachis hypogaea] | https://www.ncbi.nlm.nih.gov/protein/QHO09623.1 | NCBI Protein, QHO09623.1 |
| Chinese Academy of Agricultural Sciences | 2019 | voltage-gated hydrogen channel 1 [Benincasa hispida] | https://www.ncbi.nlm.nih.gov/protein/XP_038886538.1 | NCBI Protein, XP_038886538.1 |
| University of Veterinary Medicine Hannover | 2018 | Voltage-gated hydrogen channel 1 [Trichoplax sp. H2] | https://www.ncbi.nlm.nih.gov/protein/RDD43770.1 | NCBI Protein, RDD43770.1 |
| Fisheries and Oceans Canada | 2020 | voltage-gated hydrogen channel 1-like [Oncorhynchus keta] | https://www.ncbi.nlm.nih.gov/protein/XP_035634051.1 | NCBI Protein, XP_035634051.1 |
| Deakin University | 2017 | voltage-gated hydrogen channel 1 [Amphiprion ocellaris] | https://www.ncbi.nlm.nih.gov/protein/XP_023152539.1 | NCBI Protein, XP_023152539.1 |
| Wellcome Sanger Institute | 2019 | voltage-gated hydrogen channel 1 [Denticeps clupeoides] | https://www.ncbi.nlm.nih.gov/protein/XP_028830549.1 | NCBI Protein, XP_028830549.1 |
| Yellow Sea Fisheries Research Institute, Chinese Academy of Fishery Sciences | 2019 | voltage-gated hydrogen channel 1 [Epinephelus lanceolatus] | https://www.ncbi.nlm.nih.gov/protein/XP_033488479.1 | NCBI Protein, XP_033488479.1 |
| Wellcome Sanger Institute | 2019 | voltage-gated hydrogen channel 1 [Myripristis murdjan] | https://www.ncbi.nlm.nih.gov/protein/XP_029918050.1 | NCBI Protein, XP_029918050.1 |
| King Abdullah University of Science and technology | 2017 | voltage-gated hydrogen channel 1 [Acanthochromis polyacanthus] | https://www.ncbi.nlm.nih.gov/protein/XP_022070642.1 | NCBI Protein, XP_022070642.1 |
| BGI-SZ | 2018 | voltage-gated hydrogen channel 1-like [Tachysurus fulvidraco] | https://www.ncbi.nlm.nih.gov/protein/XP_027031192.1 | NCBI Protein, XP_027031192.1 |
| Mammalian Gene Collection Program Team | 2002 | HVCN1 protein [Homo sapiens] | https://www.ncbi.nlm.nih.gov/protein/AAH32672.1 | NCBI Protein, AAH32672.1 |
| Anon | 2011 | voltage-gated proton channel kHv1 [Karlodinium veneficum] | https://www.ncbi.nlm.nih.gov/protein/AEQ59286.1 | NCBI Protein, AEQ59286.1 |
| Okinawa Institute of Science and Technology Graduate University (OIST) | 2015 | voltage-gated hydrogen channel 1 [Lingula anatina] | https://www.ncbi.nlm.nih.gov/protein/XP_013413952.1 | NCBI Protein, XP_013413952.1 |

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
