## [Decision Letter]

**Acceptance summary:**

This elegant study describes cloning and full biophysical characterization of the first ion channel ever identified in reef-building coral species, and develops a mechanistic model for understanding regulation of these voltage-gated proton channels. It will be of interest both to marine biologists and to biophysicists studying voltage-gated proton channels in other species.

**Decision letter after peer review:**

Thank you for submitting your article "Discovery and characterization of Hv1-type proton channels in reef-building corals." for consideration by *eLife*. Your article has been reviewed by 3 peer reviewers, including László Csanády as the Reviewing Editor and Reviewer #1, and evaluation has been overseen by Kenton Swartz as the Senior Editor.

Essential Revisions (for the authors):

1. In some of the experiments, e.g., the one shown in Figure 5B, the voltage pulses are too short to allow the currents to reach steady state. This is expected to distort the derived V0.5 values (Figure 5C). Please either show current traces obtained using a protocol with longer voltage steps (especially for the human channel) or justify the use of shorter pulses (avoiding intracellular ion depletion? impracticality of inside-out patch experiments for hHv1?).

2. The suggested allosteric gating model should be discussed in the context of previously published gating models. Previous studies have suggested that the voltage sensor itself acts as a δ-pH sensor (Carmona et al., PNAS 2021, Schladt and Berger Sci. Rep. 2020), that a voltage independent but pHi dependent transition precedes opening (Villalba – Galea Biophys. J. 2014), and that the subunits within a dimeric channel allosterically influence each other (Gonzalez et al., 2010; Tombola et al., 2010).

3. The authors should add more data points to the reversal potential calculation at deltapH=2 (Figure 4B).

4. Please provide more explanation for Supplementary data – Figure 1. Based on Panel B V(0.5) is predicted to be most sensitive to deltapH in the range +2 / +3. How does that prediction agree with the saturation at deltapH > +1 observed experimentally (Figure 6B).

*Reviewer #1:*

In this paper Rangel-Yescas and colleagues identify in several Cnidarian species a gene that seems to correspond to that of the Hv1 channel in humans. Cloning and heterologous expression of the putative protein from two species of reef-building corals indeed gives rise to voltage-activated proton currents. The authors provide a careful detailed biophysical characterization of gating for the cnidarian Hv channel AmHv1. They find that coral Hv channels show much faster activation/deactivation kinetics as compared to human Hv1, but otherwise show properties similar to the latter. The science is solid. Strengths of the paper include the following: (i) AmHv1 is so far the first ion channel to be cloned from a scleractinian species. Its suggested role in coral physiology might open up novel lines of research. (ii) The authors develop a mechanistic gating model which explains all the observed gating properties, including non-linear dependence on δ-pH of current activation midpoint-voltages (V0.5). The model offers a first framework for understanding the molecular mechanisms of voltage- and proton-dependence of Hv channel gating.

1. In Figure 5B the 500 ms voltage pulse seems too short for the human channel – the currents do not reach steady state. Would the difference in V0.5 values (Figure 5C) hold up even if the pulse protocol was prolonged for the human channel? With other words: is there a true difference between steady-state activation properties, or is it only the kinetics of activation that differs for the two orthologs?

2. Do the authors have any candidate protonatable side chains in the outer and inner vestibule, in light of the pKa values of 3.4 and 7, respectively, obtained from the model fit?

3. The simplified gating model in Figure 7 is a single-subunit model. In Figure 3 the authors show evidence that AmHv1 channels dimerize. Moreover, in dimeric Hv1 channels from other species the two subunits were shown to allosterically influence each other (Gonzalez et al., 2010; Tombola et al., 2010). Maybe this fact should be mentioned in the Discussion.

*Reviewer #2:*

The authors cloned Hv1 channels from two different coral species of the genus Acropora, A. millepora and A. palmata. They found that the proteins have high sequence homology and very similar biophysical properties, despite the fact that the two organisms live in different oceans. Compared to human Hv1, the coral channels activate much more rapidly in response to membrane depolarization. The authors characterized AmHv1 in detail and used a FRET-based approach to investigate its subunit stoichiometry. Their finding is in agreement with a dimeric assembly similar to other known Hv1s. Ion selectivity and sensitivity to extracellular zinc were also investigated and found to be comparable to those of other Hv1s.

Proton currents from Hv1 channels are modulated by the pH gradient across the membrane (deltapH). The authors discovered that the deltapH dependence of the proton current from AmHv1 shows signs of saturation at values larger than one pH unit. Based on this finding, they propose an allosteric model of pH-dependent gating based on two proton binding sites, an intracellular excitatory site, and an inhibitory extracellular site. The two sites are assumed to modulate the opening transition through allosteric coupling factors. This mechanism of pH-dependent gating can have general implications when discussed in the context of alternative models in which the S4 transmembrane segment is the pH sensor.

This is a very interesting characterization of Hv1 channels from stony corals. Data and analysis convincingly support the authors' conclusions, but the following points need to be addressed:

The proposed allosteric model of pH-dependent gating is compelling, but should be discussed in the context of recent studies suggesting a mechanism in which S4 itself is the deltapH sensor (Carmona et al., PNAS 2021, Schladt and Berger Sci. Rep. 2020).

Also, there should be more discussion about how the allosteric model developed to explain AmHv1 behavior could be applied to other Hv1 channels in which the saturation of the deltapH dependence of opening is not so evident.

Supplementary data – Figure 1 needs more explanation. Panel A should specify the range of deltapH covered by the Po-V curves. If the range is the same as in the x-axis of panel B, that should be indicated. Panel B shows that the major transition in V(0.5) predicted by the model occurs in the deltapH range +2 / +3. It is difficult to understand how such a prediction can be in agreement with a saturation at deltapH > +1 observed experimentally.

In this model, the protonation/deprotonation of the intra and extracellular sites directly affect the voltage-dependent C – O transition. How does this fit with the previous suggestion that a voltage independent, but pHi dependent, transition precedes opening of other Hv1 channels (Villalba-Galea Biophys. J. 2014). Do coral Hv1 have such transition? This seems an important question to address before one can generalize the findings to Hv1 homologs like the human channel.

The ideas that coral Hv1 is involved in CaCO3 deposition and that ocean acidification could inhibit channel activation are interesting but rather speculative given that it is not known which coral cells express the channel and in which compartment or developmental stage. The discussion should be toned down or at least other possible functions mentioned.

*Reviewer #3:*

This is a very integral work, which couples several experimental methodologies in order to evidence the presence of Hv1 in reef-corals. The fact that this ancient species, the reef-corals, express Hv1, with all of its molecular and functional hallmarks, is a very interesting discovery, highlighting that Hv1 seems to be expressed in a myriad of cell-types and organisms dating as far back as the corals do. The present article is very clear, easy to read, and the conclusion are evident at the light of the data given to the reader. The authors show to have a great grasp over the tools used, be it genetics, electrophysiology or bioinformatics

1 – Although zinc is a classic inhibitor of voltage-gated proton channels Hv1, I would expect for experiments with 5-Cl^-^2GBI, which is a more specific inhibitor, in order to characterize the biophysical effect on the macroscopic currents.

2 – The authors should clarify whether the conductance computation was obtained from macroscopic OFF currents or macroscopic ON currents. It is very difficult to see in figures 4 A, 5 A and 5 B if the tail currents reach saturation. This is a very important point because it informs if the channels reached maximum conductance when the final tail current traces are overlapped, if this did not happen the maximum conductance would be underestimated which could bias any other parameter obtained by the Boltzmann fitting, e.g., the V0.5 or the Voltage dependence (q).

3 – It is debatable whether the figures (Figure 4 A and Figure 5 A) reach the steady state, but undoubtedly the figure 5 B does not reach the stationary state. This could be a problem in the choice of this specific electrophysiological recording that was chosen. I strongly suggest changing that figure and put one that reaches the steady state.

4 – When calculating the reversion potential, some points doesn't seem to have more than 1, if any, replications, which lacks real statistical relevance. In order to strengthen this beautiful result, I suggest to increase the number of this experiment.

5 – The use of a perfusion system to correct for the proton depletion classically present on Hv1 recordings was a very good idea. This could allow you to reach more depolarizing voltages without diminishing the current intensity in order to determine the precise maximum conductance thus correcting the parameters on the fitting. Moreover, this could allow for the isolation of the activation process, hence, eliminating the possible contamination of the deactivation process in the time constant calculation. If desired an instantaneous voltage protocol could allow the exclusively calculation of the deactivation process avoiding any contamination from the activation process.

---

## [Author Response]

Essential Revisions:1. In some of the experiments, e.g., the one shown in Figure 5B, the voltage pulses are too short to allow the currents to reach steady state. This is expected to distort the derived V0.5 values (Figure 5C). Please either show current traces obtained using a protocol with longer voltage steps (especially for the human channel) or justify the use of shorter pulses (avoiding intracellular ion depletion? impracticality of inside-out patch experiments for hHv1?).

The reviewers are completely right that the pulses are short, especially for the human channel. The use of shorter pulses allows for the larger depolarizations needed to activate these channels and reduce proton depletion. Regardless, we have strived to reduce the inadequacies related to the use of short pulses and have modified figure 5 as follows: We have left the short pulses for comparison between the two channels and included pulses twice as long for AmHv1 (5B) and even longer for hHv1 (5D) to reach steady state. In the case of the human channel, we have only used cells with low expression to reduce the influence of proton depletion in the longer pulses. The GVs and kinetics obtained from these longer pulses are now shown (grey and yellow triangles).

2. The suggested allosteric gating model should be discussed in the context of previously published gating models. Previous studies have suggested that the voltage sensor itself acts as a δ-pH sensor (Carmona et al., PNAS 2021, Schladt and Berger Sci. Rep. 2020), that a voltage independent but pHi dependent transition precedes opening (Villalba – Galea Biophys. J. 2014), and that the subunits within a dimeric channel allosterically influence each other (Gonzalez et al., 2010; Tombola et al., 2010).

Thank you for these suggestions. We have expanded our discussion to include these two recent publications. Our allosteric model is not in contradiction with the proposal in these papers that the voltage sensor is the pH sensor as well. The proposed binding sites in our model could be formed by the voltage sensing S4 segment and other residues. In fact, we favor this possibility. The advantage of allosteric models is that energetic arguments can be more naturally made than in phenomenological models. This discussion can be found in lines 377-386.

3. The authors should add more data points to the reversal potential calculation at deltapH=2 (Figure 4B).

Thanks for the suggestion, we have increased the number of experimental points at ΔpH = 2 and ΔpH = -1 and carried out measurements at ΔpH = 0, all of which are now included in the figure.

4. Please provide more explanation for Supplementary data – Figure 1. Based on Panel B V(0.5) is predicted to be most sensitive to deltapH in the range +2 / +3. How does that prediction agree with the saturation at deltapH > +1 observed experimentally (Figure 6B).

Please note that the model parameters that provide a good fit to our data (given in figure 6’s legend) are different from the ones used in the simulations in Supplementary data-Figure 1. We detail this further in the Discussion section. The parameters used in panel A and B are different from those in Figure 6 (all are included in the respective Figure legends). Panels C and D of the Supplementary Data-Figure 1 explore a larger range of model values. It can be seen the range of pH values over which the steeper sensitivity occurs and where saturation happens depends on the model parameters, as expected. This figure is meant to illustrate the fact that the model can be applied to channels in which pH-dependent activation happens in different ranges.

Reviewer #1:1. In Figure 5B the 500 ms voltage pulse seems too short for the human channel – the currents do not reach steady state. Would the difference in V0.5 values (Figure 5C) hold up even if the pulse protocol was prolonged for the human channel? With other words: is there a true difference between steady-state activation properties, or is it only the kinetics of activation that differs for the two orthologs?

Please see response to Essential Revision 1.

2. Do the authors have any candidate protonatable side chains in the outer and inner vestibule, in light of the pKa values of 3.4 and 7, respectively, obtained from the model fit?

The pKa values in the model are not meant to represent a single protonatable residue, likely, the proton binding site will be formed by a number of residues involved in electrostatic interactions, giving rise to a collective pKa value (please see lines 380-383). This agrees with experiments in which elimination of almost all individual protonatable residues one by one does not abolish pH-dependent gating (Ramsey, I.S., Mokrab, Y., Carvacho, I., Sands, Z.A., Sansom, M.S. and Clapham, D.E., 2010. An aqueous H^+^ permeation pathway in the voltage-gated proton channel Hv1. *Nature structural and molecular biology*, *17*(7), p.869). We are currently trying to identify to identify such an electrostatic network.

3. The simplified gating model in Figure 7 is a single-subunit model. In Figure 3 the authors show evidence that AmHv1 channels dimerize. Moreover, in dimeric Hv1 channels from other species the two subunits were shown to allosterically influence each other (Gonzalez et al., 2010; Tombola et al., 2010). Maybe this fact should be mentioned in the Discussion.

Please see our response to Essential Revisions 2.

Reviewer #2:This is a very interesting characterization of Hv1 channels from stony corals. Data and analysis convincingly support the authors' conclusions, but the following points need to be addressed:The proposed allosteric model of pH-dependent gating is compelling, but should be discussed in the context of recent studies suggesting a mechanism in which S4 itself is the deltapH sensor (Carmona et al., PNAS 2021, Schladt and Berger Sci. Rep. 2020).

Please see our response to Essential Revisions 2.

Also, there should be more discussion about how the allosteric model developed to explain AmHv1 behavior could be applied to other Hv1 channels in which the saturation of the deltapH dependence of opening is not so evident.

This is included in the discussion, but we have expanded it to make this point clear, see lines 377-386.

Supplementary data – Figure 1 needs more explanation. Panel A should specify the range of deltapH covered by the Po-V curves. If the range is the same as in the x-axis of panel B, that should be indicated. Panel B shows that the major transition in V(0.5) predicted by the model occurs in the deltapH range +2 / +3. It is difficult to understand how such a prediction can be in agreement with a saturation at deltapH > +1 observed experimentally.

In this model, the protonation/deprotonation of the intra and extracellular sites directly affect the voltage-dependent C – O transition. How does this fit with the previous suggestion that a voltage independent, but pHi dependent, transition precedes opening of other Hv1 channels (Villalba-Galea Biophys. J. 2014). Do coral Hv1channels have such transition? This seems an important question to address before one can generalize the findings to Hv1 homologs like the human channel.

We explain this further in the Discussion section. The parameters used in panel A and B are different from those in Figure 6 (all are included in the respective Figure legends). Panels C and D of the Supplementary Data-Figure 1 explore a larger range of model values. It can be seen the range of pH values over which the steeper sensitivity occurs and where saturation happens depends on the model parameters, as expected. This figure is meant to illustrate the fact that the model can be applied to channels in which pH-dependent activation happens in different ranges.

Our experiments do not yet distinguish between voltage-dependent and independent transitions and how they fit into an allosteric gating model. In the future we plan to expand the model and compare the observed kinetics with predictions from this and other types of models.

In this model, the protonation/deprotonation of the intra and extracellular sites directly affect the voltage-dependent C – O transition. How does this fit with the previous suggestion that a voltage independent, but pHi dependent, transition precedes opening of other Hv1 channels (Villalba-Galea Biophys. J. 2014). Do coral Hv1 have such transition? This seems an important question to address before one can generalize the findings to Hv1 homologs like the human channel.The ideas that coral Hv1 is involved in CaCO3 deposition and that ocean acidification could inhibit channel activation are interesting but rather speculative given that it is not known which coral cells express the channel and in which compartment or developmental stage. The discussion should be toned down or at least other possible functions mentioned.

Thanks for this suggestion. We have toned down this part of the discussion and mentioned the possible role of these channels in cell pH regulation (lines 419-420 and 429-432).

Reviewer #3:This is a very integral work, which couples several experimental methodologies in order to evidence the presence of Hv1 in reef-corals. The fact that this ancient species, the reef-corals, express Hv1, with all of its molecular and functional hallmarks, is a very interesting discovery, highlighting that Hv1 seems to be expressed in a myriad of cell-types and organisms dating as far back as the corals do. The present article is very clear, easy to read, and the conclusion are evident at the light of the data given to the reader. The authors show to have a great grasp over the tools used, be it genetics, electrophysiology or bioinformatics.

We are grateful for the reviewer’s assessment of our manuscript. Below we have responded to the specific reviewer’s concerns.

1 – Although zinc is a classic inhibitor of voltage-gated proton channels Hv1, I would expect for experiments with 5-Cl^-^2GBI, which is a more specific inhibitor, in order to characterize the biophysical effect on the macroscopic currents.

This is an excellent suggestion and we might revisit block by 5-Cl^-^2GBI in the future. In this work we tested Zn^2+^ because is the most widely characterized blocker of Hv1 channels and because its potential effects in coral biology through proton channel block.

2 – The authors should clarify whether the conductance computation was obtained from macroscopic OFF currents or macroscopic ON currents. It is very difficult to see in figures 4 A, 5 A and 5 B if the tail currents reach saturation. This is a very important point because it informs if the channels reached maximum conductance when the final tail current traces are overlapped, if this did not happen the maximum conductance would be underestimated which could bias any other parameter obtained by the Boltzmann fitting, e.g., the V0.5 or the Voltage dependence (q).

Thanks for pointing this out. We measured conductance from the steady state currents and not from tail currents. The original short pulses we used, while maybe sufficient for the faster AmHv1 channel are definitely not adequate for the human channel, as was pointed out by the reviewers. We have now included analysis of the conductance and kinetics from longer stimuli for bot channels to ensure that the steady state is reached, and conductance is not underestimated (Figure 5).

Tail currents were not used because, especially for the slower human channel, p/n subtraction is not plausible, and the tail current is not reliably measured in the presence of the large capacity transient.

3 – It is debatable whether the figures (Figure 4 A and Figure 5 A) reach the steady state, but undoubtedly the figure 5 B does not reach the stationary state. This could be a problem in the choice of this specific electrophysiological recording that was chosen. I strongly suggest changing that figure and put one that reaches the steady state.

Please see response to Essential Revision 1.

4 – When calculating the reversion potential, some points doesn't seem to have more than 1, if any, replications, which lacks real statistical relevance. In order to strengthen this beautiful result, I suggest to increase the number of this experiment.

Please see response to Essential Revision 3.

5 – The use of a perfusion system to correct for the proton depletion classically present on Hv1 recordings was a very good idea. This could allow you to reach more depolarizing voltages without diminishing the current intensity in order to determine the precise maximum conductance thus correcting the parameters on the fitting. Moreover, this could allow for the isolation of the activation process, hence, eliminating the possible contamination of the deactivation process in the time constant calculation. If desired an instantaneous voltage protocol could allow the exclusively calculation of the deactivation process avoiding any contamination from the activation process.

Thanks for the comment. In the future, we plan to continue using perfusion protocols combined with inside-out recordings to help reduce the influence of proton depletion and obtain recordings of processes that can be confounded by ion accumulation/depletion, such as tail currents at extreme voltages.